# Whole-cell modeling in yeast predicts compartment-specific proteome constraints that drive metabolic strategies

Ibrahim E. Elsemman[1,7,8], Angelica Rodriguez Prado[2,3,8], Pranas Grigaitis[2,8], Manuel Garcia Albornoz[4], Victoria Harman[5], Stephen W. Holman[5], Johan van Heerden[2], Frank J. Bruggeman[2], Mark M. M. Bisschops[3], Nikolaus Sonnenschein[1], Simon Hubbard[4], Rob Beynon[5], Pascale Daran-Lapujade[3], Jens Nielsen[1,6✉] & Bas Teusink[2✉]

When conditions change, unicellular organisms rewire their metabolism to sustain cell maintenance and cellular growth. Such rewiring may be understood as resource re-allocation under cellular constraints. Eukaryal cells contain metabolically active organelles such as mitochondria, competing for cytosolic space and resources, and the nature of the relevant cellular constraints remain to be determined for such cells. Here, we present a comprehensive metabolic model of the yeast cell, based on its full metabolic reaction network extended with protein synthesis and degradation reactions. The model predicts metabolic fluxes and corresponding protein expression by constraining compartment-specific protein pools and maximising growth rate. Comparing model predictions with quantitative experimental data suggests that under glucose limitation, a mitochondrial constraint limits growth at the onset of ethanol formation—known as the Crabtree effect. Under sugar excess, however, a constraint on total cytosolic volume dictates overflow metabolism. Our comprehensive model thus identifies condition-dependent and compartment-specific constraints that can explain metabolic strategies and protein expression profiles from growth rate optimisation, providing a framework to understand metabolic adaptation in eukaryal cells.

[1] Novo Nordisk Foundation Center for Biosustainability, Technical University of Denmark, DK2800 Lyngby, Denmark. [2] Systems Biology Lab, Amsterdam Institute of Molecular and Life Sciences, Vrije Universiteit Amsterdam, Amsterdam, The Netherlands. [3] Department of Industrial Microbiology, Technical University Delft, Delft, The Netherlands. [4] Division of Evolution & Genomic Sciences, University of Manchester, Manchester, UK. [5] Institute of Systems, Molecular and Integrative Biology, University of Liverpool, Liverpool, UK. [6] Department of Biology and Biological Engineering, Chalmers University of Technology, SE41296 Gothenburg, Sweden. [7] Present address: Department of Information Systems, Faculty of Computers and Information, Assiut University, Assiut, Egypt. [8] These authors contributed equally: Ibrahim E. Elsemman, Angelica Rodriguez Prado, Pranas Grigaitis. ✉email: nielsenj@chalmers.se; b.teusink@vu.nl

Macromolecular synthesis and energy conservation by metabolism underlies cellular maintenance, growth and fitness. Unicellular organisms such as yeasts generally display a great variety of metabolic strategies that lead to competitive fitness across conditions[1]. The associated reprogramming of metabolism between such metabolic strategies is of key interest in biotechnology and biomedical research.

One well-known example is overflow metabolism in which under aerobic conditions not all substrate is fully oxidised but secreted as by-products. In cancer cells, it is referred to as the Warburg effect: enhanced glycolytic activity with lactate as byproduct at the expense of respiration[2]. The same phenomenon is known as the Crabtree effect in *Saccharomyces cerevisiae* (Baker's yeast)[3]. At sugar limitation yeast respires glucose completely to $CO_2$; at sugar excess it displays respirofermentative metabolism, where respiration is combined with ethanol formation (alcoholic fermentation). The extent to which these two metabolic strategies are used can be titrated in a glucose-limited chemostat: at a specific critical dilution (=growth) rate, ethanol formation starts and increases linearly with growth rate[4]. Other microorganisms show similar behaviour[5]: for example, *E. coli* produces acetate at higher growth rates at the expense of respiration[6].

In the last decade, a theoretical framework has been developed that can explain why cells shift metabolic strategies upon environmental or gene-expression perturbations[5,7–10]. In essence, it is based on the catalytic benefits of proteins and their associated costs[11]. These costs comprise competition for resources such as building blocks, energy and synthesis machineries, and for space in cellular compartments. Two key features of this resource allocation paradigm can explain metabolic adaptations. First, cellular compartments can become full when they are fully occupied with (maximally) active proteins, such that an increase in one protein has to come at the expense of another. This was postulated as a phenomenological rule based on experimental observations[12], but also follows naturally from growth-rate maximisation[13]. Second, cells allocate their limited resources for protein synthesis according to their demands[14,15]. Consequently, fractions of needed proteins vary with growth rate within compartments whose protein content is bounded, and this can lead to active proteome constraints related to full compartments.

Within this framework, the onset of overflow metabolism was explained by the smaller protein cost of generating ATP through fermentation than respiratory pathways[6,7]; this becomes important at fast growth when biosynthesis and ribosome demands are high and thus require large proteome fractions. Earlier work suggests that the proteome-constrained resource allocation paradigm, which was largely developed for *E. coli*, may also be a powerful perspective for regulation of eukaryal yeast metabolism, such as ribosome biosynthesis[16], and growth on different sugars[17]. However, a key feature of the metabolism of a eukaryal cell is the presence of metabolically active organelles, most prominently mitochondria. Each organelle introduces two new compartments (intra-organellar space and membrane), and how these compartments impact adaptation of metabolism, and which compartments become limiting under different conditions, is an open question.

Moreover, despite the wealth of experimental data on *Saccharomyces cerevisiae*, a comprehensive, quantitative, data set in which growth rate is systematically varied and both fluxes and protein expression levels are measured, which are needed to validate resource allocation predictions, are still rare (see, however, some recent studies[16,18]). Here, we present such data sets and, in parallel developed, detailed and comprehensive, compartmentalised and quantitative model of metabolism and protein synthesis of yeast. The model can compute the costs and benefits of protein expression and translocation; It can be used to interpret or predict experimentally determined changes in growth rate, (minimal) protein expression and metabolic fluxes as a result of growth rate optimisation through resource allocation into different, compartmentalised, proteome fractions. Comparison of the model predictions with the data gives unprecedented insight into our physiological understanding of this important model organism.

## Results

**Construction of a comprehensive proteome-constrained yeast model.** We extended an existing[19] metabolic genome-scale metabolic model of yeast (GEM) by coupling metabolic fluxes to the synthesis of the catalysing enzyme and added constraints on protein concentrations, expressed as protein fractions of the total proteome (Fig. 1a). We refer to the resulting model as proteome-constrained Yeast (pcYeast). Earlier GEM-based approaches exist that incorporate resource allocation, and for yeast these considered constraints on enzyme activities and total protein content[17,20–23], whereas for *E. coli* constraints and reactions associated with transcription and translation were added[9]. Others considered membrane-area constraints and limitations of protein allocation to specific pathways[8,24]. We combined all these extensions (see Supplementary Notes 1–6 for detailed information) to make pcYeast: a next-generation yeast GEM and computable knowledge base that incorporates protein expression, translation, folding, translocation and degradation at genome-scale for a compartmentalised, eukaryal, organism. In our current model, we consider the protein compartments most relevant for central metabolism: plasma membrane, cytosol, mitochondrion and mitochondrial membrane. Other cell compartments such as the nucleus or endoplasmic reticulum are not (yet) specified explicitly - but do occupy volume in the cytosol.

The cellular proteome was divided into metabolically active, ribosomal, and unspecified (UP) proteins. The UP fraction is cytosolic, has an average amino acid composition and is added to always maintain a constant protein density in the cytosol. It has a minimum expression level estimated from the experimental proteomics data (Supplementary Fig. 1, Supplementary Note 2). The minimal UP fraction represents growth-rate independent structural, signalling and household proteins. Higher expression of UP than minimal represent both unspecified anticipatory proteins, or metabolic proteins that do not carry flux – including the unsaturated fraction of flux-carrying enzymes, as we will explain.

Metabolic enzymes are assigned to a specific compartment, either cytosol, plasma membrane, mitochondrial matrix or inner-mitochondrial membrane; Mitochondrial proteins require additional protein transport complexes[25]. For each protein, we comprehensively modelled synthesis and degradation processes, which are responsible for the largest fraction of cellular energy usage. Our model includes 1523 proteins whose life cycles are described by 16,304 reactions that include translation initiation, elongation and termination factors, ribosomal assembly factors, protein-specific folding by chaperones and degradation reactions, as well as 5'UTR-length dependent energetic costs for translation initiation (Table 1, Supplementary Note 2).

We applied three classes of constraints that couple metabolic fluxes and peptides synthesis rates (Fig. 1b and Supplementary Note 2 for details). The *enzyme capacity constraint* sets the minimal enzyme synthesis rate required to achieve a certain metabolic flux. Thus, all metabolically-active proteins are modelled to work at their maximal rate and are minimally expressed in the model; the unsaturated fraction of flux-carrying enzymes is represented by UP, the unspecified protein that is used

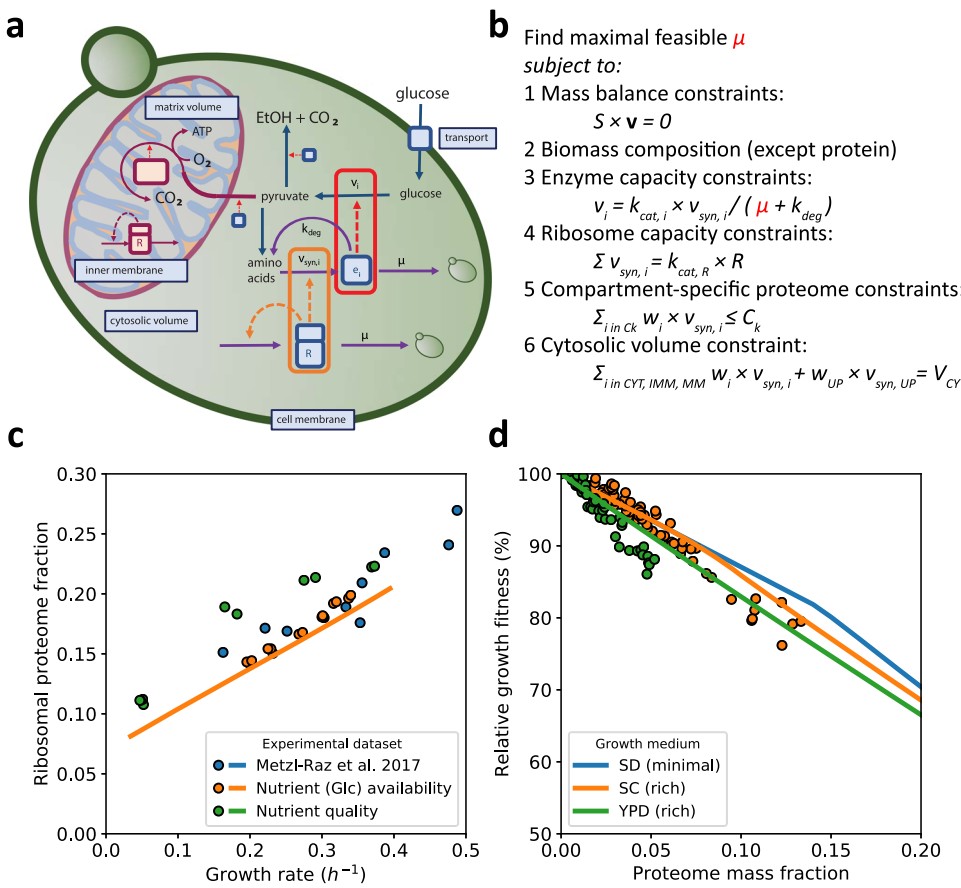

**Fig. 1 pcYeast model formulation and calibration of protein synthesis parameters. a** A schematic overview of reactions in the model, their interdependence and constraints. Metabolic reactions $v_i$ are proportional to enzyme concentrations $e_i$ that are synthesised at rate $v_{syn,i}$ by the ribosomes $R$. Each protein can be degraded with rate $v_{deg,i} = k_{deg} \cdot e_i$ or diluted by growth rate $v_{dil,i} = \mu \cdot e_i$. Compartment-specific constraints are indicated in the light-blue boxes. **b** Optimisation problem with the key constraints, including (1) steady-state mass balances; (2) production of biomass components such as DNA, lipids, cell wall and polysaccharides. Proteins are excluded as their synthesis rates are optimisation variables (3) enzyme capacity constraints that couple metabolic flux to catalytic rate $k_{cat,i}$ and the enzyme level, whose value at steady state is determined by its synthesis rate. Note we use equalities and hence enzymes work at their maximal rate and minimal required protein levels are computed; (4) ribosome capacity that defines an upper bound for protein synthesis rate; (5) compartment-specific proteome constraints that define the maximal concentration of proteins that can be contained in that compartment, with $w_i$ the specific volume or area of protein $i$; (6) a cytosolic protein density constraint that has the same function as that of proteome constraints, but whose equality forces the cell to fill up any vacant proteome space with unspecified protein UP. **c** Growth rate was varied through sugar type (trehalose, galactose, maltose, glucose) or glucose concentration, and ribosomal protein fraction was determined by proteomics. The translation rate was calibrated on the literature data (Supplementary Notes 6). **d** Impact of mCherry protein overexpression on growth rate. Symbols show experimental data[64], solid lines show model predictions based on glucose minimal (SD) medium or rich SC/YPD media. Model predictions were obtained by varying the proteome mass fraction, occupied by mCherry, and determining the maximal predicted growth rate at each value of the mass fraction. The relative growth fitness represents the ratio between the growth rate at certain mCherry expression level vs. the unperturbed state (no mCherry expression). Source data for panels **c** and **d** are provided as a Source Data file.

to maintain protein density. In this way we prevent choices about unknown regulatory and kinetic mechanisms that may affect the activity of enzymes; rather we use the deviation between predicted minimal and measured actual protein expression levels to indicate the level of saturation of each enzyme. The total enzyme synthesis rate is constrained by the abundance of ribosomes through a *ribosome capacity constraint*, for both cytosol and mitochondria. Finally, we added *compartment-specific constraints* on the proteome, for the cytosol, the plasma membrane, and the mitochondrial matrix and inner membrane, (Fig. 1b). The values for these constraints are based on independent literature data or were fitted to experimental data (as explicified in Supplementary Notes 2 and 3) and the values are either fixed or growth-rate dependent, depending on the nature of the constraint.

The steady-state metabolite balances, the enzyme synthesis and degradation balances, and the compartment-specific proteome constraints together specify a linear program with its fluxes as

optimisation variables, provided the growth rate is treated as a parameter. We use a binary search algorithm to find the maximum growth rate where the linear program is still feasible, and a marginal increase in the growth rate would result in an infeasible linear programming problem. The model returns all the flux values associated with the maximal feasible growth rate. It should be noted that the structure of the pcYeast model is strain-independent: this allows subsequent calibration of the model to accommodate and account for differences in cell physiology and metabolism, inherent to any specific strain of *S. cerevisiae*.

**Calibrating the model against experimental data.** We performed a series of experiments, using a wild-type *S. cerevisiae* strain CEN.PK 113-7D, for collection of high-quality datasets of fluxes and protein levels, used either as model input or for comparison with model predictions. We used glucose-limited

**Table 1 Statistics of the pcYeast model.**

| Process/compartment | # of reactions | # of proteins |
|---|---|---|
| Total | 24422 | 1520 |
| Metabolic network | 5774 | 913 |
| from Yeast7.6 | 5738 | 909 |
| manually added metabolic reactions | 36 | 4 |
| Cytoplasm | 2349 | 778 |
| Plasma membrane | 529 | 114 |
| Mitochondria | 1089 | 272 |
| Endomembrane system | 2127 | 133 |
| Metabolic complex formation, disassembly, dilution | 2787 | – |
| tRNA turnover and modification | 2194 | 56 |
| Protein synthesis and turnover | 13312 | 403 |
| Cytoplasmic translation | 1512 | 138 |
| Mitochondrial translation | 8 | 89 |
| Protein folding | 1515 | 31 |
| Protein degradation | 1607 | 42 |
| Protein misfolding, refolding | 6061 | 73 |
| Protein transport | 1324 | 30 |
| Protein dilution by growth | 1285 | – |
| Formation of macromolecular complexes | 355 | 196 |

continuous cultures operated at dilution rates close to the critical dilution rate for ethanol formation, to capture proteome change upon the onset of overflow metabolism. Additionally, we varied the growth rate in pH-controlled batch experiments, either with different sugar quality or through translation inhibition. We measured fluxes, including $O_2$ and $CO_2$ fluxes (Supplementary Data 1), which combined with biomass measurements, allowed to estimate the so-called maintenance parameters, i.e., ATP usage that is not explicitly accounted for in the model (Supplementary Note 2). Label-free proteome quantification allowed us to reliably estimate proteome fractions of around 3000 of the 6000 proteins (Supplementary Data 2, 3, and 4).

Parameters associated with translation strongly affected our model outcomes, and we used published quantitative proteomics data[16] to estimate parameters for protein translation, such as the elongation rate (Supplementary Notes 2 and 6). Following experimental reports we assumed a constant inactive fraction of ribosomes and a fixed saturation of the actively translating ribosomes[16,26] and were able to describe the growth-rate dependent ribosome mass fraction with the model (Fig. 1c). As evidence for correctly capturing the costs of protein synthesis, we correctly predicted the effect of over-expressing mCherry, an unneeded, gratuitous protein, on the specific growth rate (Fig. 1d).

**The model predicts shifts in metabolic strategies**. We subsequently used the model to analyse yeast's physiological response to different levels of glucose availability. Traditional Flux Balance Analysis computes continuous chemostat cultures by minimising glucose uptake rate at fixed growth (=dilution) rate[27]. Here, we simulated glucose availability by varying the degree of saturation of the glucose transporter. We needed to constrain the maximal expression level of the glucose transport system based on literature data (Supplementary Note 2), as leaving expression free to occupy available membrane space led to unrealistically high expression levels and overestimation of growth rate at low glucose levels. At each saturation level, we computed the maximal feasible growth rate and compared model predictions with published data[28], and with data from our glucose-limited chemostat cultures (growth rates between $0.2–0.34\,h^{-1}$). We also included our data

from batch cultures on glucose (growth rates $0.37–0.39\,h^{-1}$) and on trehalose; Trehalose is a disaccharide of two glucose molecules, hydrolysed extralullarly[29], thus providing slow release of glucose that supports low growth rates.

As in the chemostat, the specific growth rate is equal to the dilution rate, the maximal feasible growth rate that the model predicted can be directly compared to the experimental data (Fig. 2a, c, d). The (residual) glucose concentrations were calculated from documented (high) affinity of the transporters, which is close to 1 mM[30]. The resulting relationship between growth rate and residual glucose concentration fit experimental data very well (Fig. 2a), validating our expectation that we could ignore glucose efflux from the cells due to minute levels of intracellular glucose[31] (see Supplementary Note 2 for details). Increasing glucose transporter saturation increased predicted growth rate, and the effect saturated (Fig. 2b), suggesting that at maximal growth rate further increase in glucose availability has little impact. Predicted biomass yield (Fig. 2c) and fluxes (Fig. 2d) corresponded well with the experimental data, as did the intracellular flux ratios from previously published $^{13}C$-labelling flux analysis at three specific growth rates in glucose-limited chemostat cultures (Supplementary Fig. 2). In particular, the model predicted a maximal oxygen consumption rate at dilution rates higher than $0.28\,h^{-1}$, at the onset of ethanol formation. Above $0.35\,h^{-1}$, this rate rapidly drops to the low level that is observed under glucose excess (batch) conditions. We conclude that the model can adequately predict the changes in metabolic fluxes when the growth rate is varied through the availability of glucose.

**Changes in metabolic strategies are the result of proteome constraints**. We used pcYeast to identify the active proteome constraints, i.e., the protein pools that limit growth rate, because, according to resource allocation theory, the number of active proteome constraints determines the maximal number of independent metabolic behaviours that are possible in optimal states[5,13]. For this, we computed the occupancy of each protein pool: a pool that is fully occupied is indicative of an active constraint. At low growth rates, below $0.28\,h^{-1}$, the glucose transporter was the only proteome pool that is fully occupied (Fig. 2e). With only glucose uptake as active constraint, pure respiration is the single optimal strategy. At the onset of ethanol formation a second metabolic mode started to carry flux (for formal computation of these modes and the concomitant theory, see Supplementary Note 4), and thus a second constraint must have become active. Indeed, at this growth rate the occupancy of the inner-mitochondrial membrane became maximal (Fig. 2e). Thus, the model suggests that under glucose-limited chemostat conditions, the onset of ethanol formation is caused by a limit of the mitochondrial membrane space, and hence the amount of proteins that yeast can maximally express in this compartment.

At a growth rate of $0.35\,h^{-1}$ we found that the unspecified protein level reached its minimal value (Fig. 2e), equivalent to the cytosol being completely filled with maximally active proteins. Further growth rate increase requires higher ribosomes and biosynthetic protein fractions, which now has to come at the expense of the least proteome-efficient pathway. The model confirmed earlier calculations[32] that respiration is less proteome efficient than fermentation (Supplementary Fig. 3) and respiration is therefore replaced by fermentation. The model suggested, therefore, that at growth rate above $0.35\,h^{-1}$ the second growth-limiting constraint was shifted from the mitochondrial proteome to the cytosolic proteome. Thus, the metabolic changes in the model, when growth rate and thus metabolic fluxes increase, are dictated by the filling up of different cellular compartments with

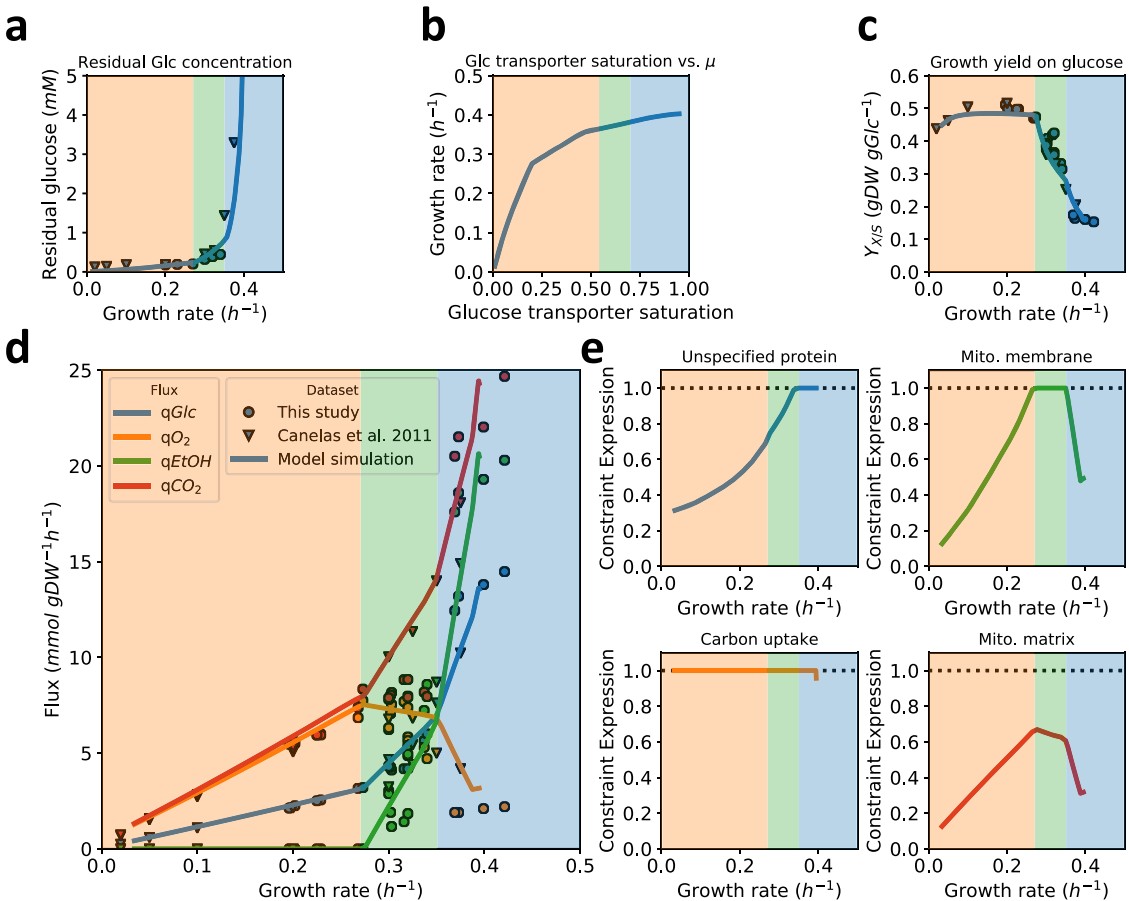

**Fig. 2 Predicted and measured physiological response of S. cerevisiae CEN.PK as a function of glucose availability.** Predicted (lines) and measured (symbols) physiological parameters and fluxes of *S. cerevisiae* CEN.PK strain **a** Measured (symbols) and predicted (line) residual glucose concentrations as a function of growth rate. The latter was calculated based on published affinity for glucose and assuming negligible intracellular glucose under these conditions. Note that this resembles a Monod growth curve but with the dependent and independent axis swapped, as we control growth rate in a chemostat. **b** Maximal feasible growth rates of the model as a function of the glucose transporter saturation. **c** Measured (symbols) and predicted biomass yield on glucose. **d** Experimental fluxes from glucose-limited chemostats at different dilution rates and from two batch experiments: excess trehalose (which mimics glucose limitation at low dilution rate[29]) and excess glucose at the highest growth rate. The lines are model predictions; **e** Computed proteome occupancy of different constrained protein pools. A fraction of 1 means that the compartment is full with metabolically actively proteins and constrains the growth rate at that condition. The shading of the different growth regimes is based on the (latest) constraint, actively limiting growth, referring to Panel (**e**). Source data are provided as a Source Data file.

active protein, unique for an eukaryal cell. The level of detail in our model to suggest the condition-dependent, active, protein-concentration constraints belonging to different compartments has so far not been provided by any other model.

**Proteomics data validates model predictions.** We subsequently measured protein levels with quantitative proteomics and compared them to the minimal protein levels that the model predicted to be needed to support metabolic flux. Since we compute minimal levels as if all the enzymes worked at their maximal rate, we expected to underestimate most proteome fractions. Especially at lower growth rates where nutrient limitation is most severe, one can expect lower average enzyme saturation, and indeed we observed larger deviations between predicted minimal protein levels and measured protein fractions at low growth rates (Fig. 3a). The difference between the predicted minimal level and the data may be interpreted as a proxy for the average saturation of enzymes. In terms of protein synthesis costs, the difference between the experimentally measured enzyme expression and the predicted minimal expression level, however, are covered by the

expression of the UP. We see an overall tendency that the saturation of enzymes increases with growth rate (Supplementary Fig. 4). This is most prominent for the glycolytic pathway; also for amino acid biosynthesis, the protein expression is higher than expected based on metabolic activity, indicating also here a substantial undersaturation of the enzymes, as observed before for bacteria such as *E. coli*[33] and *L. lactis*[34]. We find similar patterns for other biosynthetic pathways, except for lipids (Supplementary Fig. 5).

For mitochondrial proteins involved in the citric acid cycle and respiration, however, we found that predicted minimal proteome fractions were very close to the measured ones (Fig. 3a). Unless $k_{cat}$ values of mitochondrial enzymes are systematically underestimated, this indicates that mitochondrial proteins work at higher average saturation than cytosolic proteins - and seemingly close to their maximal capacity. Regardless of absolute numbers, the saturation of the mitochondria seems rather constant, suggesting that yeast tunes the total amount of mitochondria, rather than make excess (hence subsaturated) mitochondria, at least under these conditions. This may make sense, given the extra costs of mitochondrial components such as membranes,

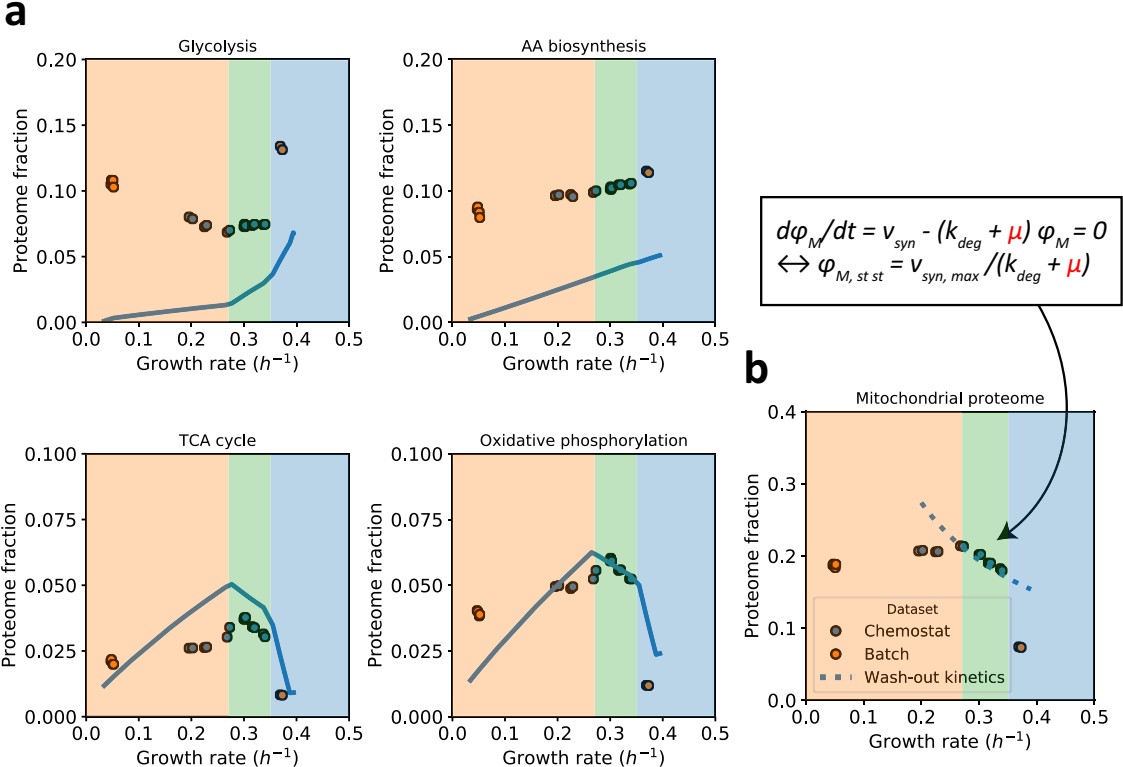

**Fig. 3 Proteomics data of selected pathways as a function of glucose availability.** Blue symbols are glucose-limited chemostat data; orange symbols are controlled batch experiments with excess trehalose (lowest growth rate) or glucose (highest growth rate) **a** Comparison of predicted minimal proteome fractions to sustain growth with the experimentally determined proteome fraction for four pathways. The ratio between the two represents an estimate of the saturation level of the constituent enzymes. Lines represent the model; experimental data are symbols. **b** Decay of steady-state mitochondrial protein fraction with growth rate at onset of ethanol formation suggests a maximal rate of mitochondrial biosynthesis $v_{syn,max}$. The shading of the different growth regimes is based on the (latest) constraint, actively limiting growth, referring to Fig. 2e. Individual proteins in panel **a** were mapped to metabolic pathways using a manually-curated pathway annotation file (Supplementary Data 5). Source data are provided as a Source Data file.

and for protein translocation of host-derived proteins during mitochondrial biogenesis, which competes for membrane space with respiratory proteins.

Upon closer inspection, we observed that at the onset of ethanol formation the total mitochondrial protein fraction started to decrease (Fig. 3b). The observed decay follows the theoretical dilution-by-growth kinetics if at that point the rate of mitochondrial biosynthesis has reached a maximum (Fig. 3b). Thus, the data suggest that the rate of mitochondrial biogenesis, rather than the maximal mitochondrial membrane area currently used by the model, may reach the host's maximal capacity at the onset of ethanol formation. When we zoom in on the mitochondrial proteome, we find that the mitochondrial ribosome fraction increased as a funtion of growth rate, and also other proteins re-allocated (Supplementary Fig. 6). Indeed, mitochondria are self-replicating entities abiding to the same resource allocation principles as the host, which even includes selection for fast replication - but obviously severely dictated by the proteins the host provides. More data related to the mitochondrial biosynthetic processes, such as mitochondrial ribosomal capacity and protein import machinery would be required to predict the maximal mitochondrial growth rate from first principles, which is outside the scope of this study. Nonetheless, the distinct changes of mitochondrial proteins at the critical dilution rate are consistent with the model prediction that a mitochondrial constraint is responsible for the onset of ethanol formation under glucose-limited conditions.

**Constraints and fluxes under sugar excess conditions**. We then varied the growth rate (between $0.05\,h^{-1}$ and $0.4\,h^{-1}$) by providing different sugars, i.e., trehalose, galactose, maltose and glucose during batch cultivation. Ethanol production was already observed on galactose, already at a growth rate of $0.16\,h^{-1}$ so at a much lower growth rate than the critical growth rate of $0.28\,h^{-1}$ under glucose-limited growth (Fig. 4a). Maltose showed intermediate growth rate and fluxes. Initial model simulations with a naïve model using the reported catalytic rates of the transporters and catabolic enzymes involved in galactose and maltose metabolism, however, resulted in predicted growth rates and fluxes not far from growth on glucose. This suggests that there are additional cost factors that were not included in the model, and or that *Saccharomyces cerevisiae* is not as well adapted to these sugars.

We, therefore, used the model as data analysis tool to estimate possible changes in parameters that fit the observed growth rate and corresponding fluxes (see Supplementary Notes 3 and 5 for details, summary provided in Table 2). Such parameter changes may be interpreted as costs for suboptimal metabolism of carbon sources other than glucose. The onset of ethanol formation at a growth rate of $0.16\,h^{-1}$ required a combination of changes in both sugar uptake and the intracellular proteome (through the minimal UP fraction constraint): a lower sugar uptake capacity alone would be identical to lowering saturation of the transporter as was done for glucose (Fig. 2), and pure respiration would have been found at $0.16\,h^{-1}$. Conversely, only an increase in minimal UP would have resulted in a proportional flux decrease

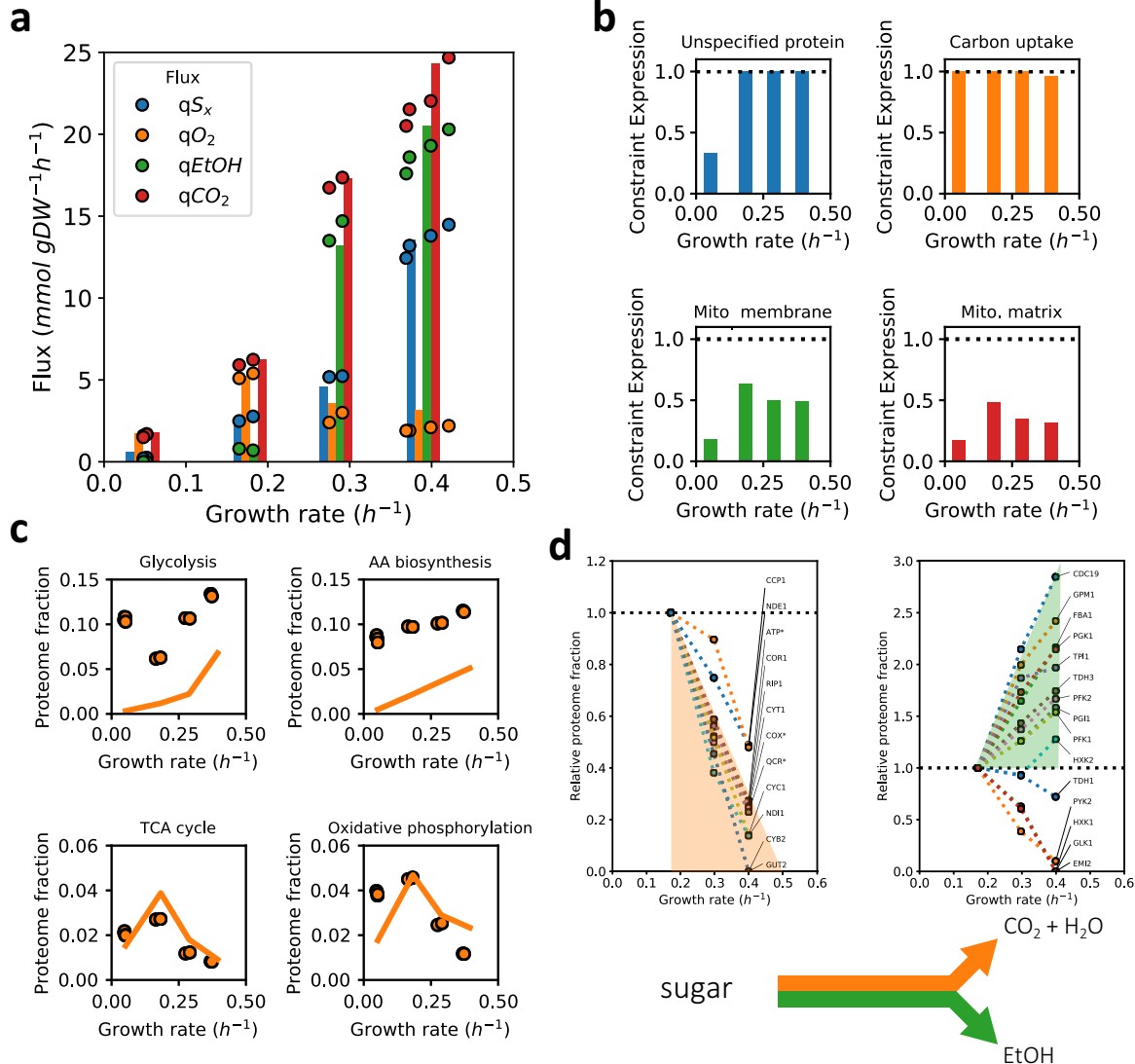

**Fig. 4 Predicted and measured physiology of _S. cerevisiae_ CEN.PK strain in sugar batch cultivations.** Model predictions, fluxes and protein levels plotted as a function of growth rate during hexose sugar excess conditions (in the order: trehalose, galactose, maltose, glucose) **a** Fluxes of sugar consumption, oxygen consumption and ethanol production. Circles are experimental data, bar plots indicate model predictions (of both the growth rate and fluxes); **b** Predicted active constraints under the different sugar excess conditions as predicted by the mode (see legend of Fig. 2 for details). **c** Comparison of predicted minimally needed proteome fractions with experimentally determined ones suggests differences in saturation level between pathways. Lines represent the model, experimental data are circles; **d** Linearity of the expression of individual enzymes in glycolysis (right) and respiration (left) with growth rate suggests trading in of respiratory protein for fermentative protein. Asterixes indicate aggregated proteome fractions instead of fractions of individual proteins. The respiratory proteins converge at $0.474 \pm 0.0002\,h^{-1}$. Shading in top plots of panel (**d**) highlights the common trend of individual protein abundance, corresponding to the end-products in the scheme on the bottom. Individual proteins in panels **c** and **d** were mapped to metabolic pathways using a manually-curated pathway annotation file (Supplementary Data 5). Source data are provided as a Source Data file.

**Table 2 Changes to the parameters for simulating sugar excess conditions. NGAM is non-growth related ATP maintenance.**

| Growth condition | Unit | Glucose (naïve) | Galactose | Maltose |
|---|---|---|---|---|
| Maximal hexose transporter area | $\mu m^2/cell$ | 7.5 | 3.0 | 3.5 |
| Carbon-related NGAM | $mmol/gDW/h$ | 0.0 | 3.0 | 0.0 |
| Minimal UP fraction | $g\ UP/g\ protein$ | 0.245 | 0.49 | 0.34 |

that we also found with mCherry overexpression (or translation inhibition, Supplementary Fig. 7), and more ethanol were to be found.

We had to decrease the maximal galactose uptake rate by a factor of 2.5 compared to glucose. Furthermore, an increase in

minimal UP fraction was needed, to 0.49 g/g protein. To fit all fluxes optimally, we also required additional energetic costs (see Supplementary Note 5), whose mechanistic underpinning remains to be explored but may be related to the reported toxicity of galactose intermediates[35]. Such a change in energetic

costs were not needed to describe the data for growth on maltose: only a change in the maltose uptake rate and minimal UP fraction (of 0.34 g/g protein) were required to achieve good fit.

For maltose, a disaccharide of glucose, the reason for the required parameter changes is not clear. Only a maltose proton-symporter and a maltase protein distinguishes it from growth on glucose. The transport expression may be tightly regulated as very high maltose uptake rates can result in substrate-accelerated death[36]. For galactose, the toxicity of its intermediates[35] results in an evolutionary trade-off with growth on glucose[37]; on galactose yeast cells appear to be still prepared for growth on glucose, which may prevent them from optimal expression of proteins on galactose, as shown by expression titration experiments[38]. Indeed, laboratory evolution experiments on galactose select mutations in Ras/cAMP signalling and adapted strains show increased growth rates and concomitant increased ethanol fluxes[37]. Interestingly, the direction of change points to the optimal behaviour predicted by the initial naïve model, suggesting that the pcYeast model may aid in predicting the direction of evolutionary change during laboratory evolution experiments (Supplementary Fig. 8).

With the updated parameters, we identified for both sugars that the active constraints limiting growth were the sugar transport expression and the minimal UP fraction constraint (Fig. 4d, Supplementary Note 5, Supplementary Fig. 9). These active constraints explain ethanol formation during growth on galactose even though the growth rate is lower than the critical dilution rate on glucose.

**Proteomics data on sugar excess shows re-allocation of metabolic strategies.** If growth rate is actively constrained by the cytosolic proteome under galactose, maltose and glucose excess conditions, it implies that all cytosolic proteins work at their maximum activity, and changes in flux must be brought about by changes in protein level. We, therefore, turned to proteomics again. Comparing the minimal levels of the model with experimental data, we find again that mitochondrial proteins for the TCA cycle and respiration are very similar to the predicted minimal levels required to sustain flux (Fig. 4c). Cytosolic proteins were underestimated - even at sugar excess conditions. (Note, however, that the expected maximal attainable activity is not likely at the maximal rate in the forward direction as product inhibition is inevitable in a chain of enzymes.)

More indicative of 'a full cytosol' is that at the onset of ethanol formation (at galactose growth rate and higher), we find evidence for proportional relationships between protein and flux for high-flux carrying, pathway-grouped, proteins as a function of growth rate (Fig. 4c). This is observed even down to the individual protein level (also involving changes in expressed isozymes), as illustrated for glycolytic and respiratory proteins in Fig. 4d. This implies that under these conditions, enzyme saturation was constant (and maximal, we expect) and changes in flux could only be brought about by corresponding changes in enzyme levels. This data illustrates how mitochondrial proteins are being traded in for glycolytic proteins needed for an enhanced fermentation and growth rate. It also confirms the model's prediction that the cytosolic proteome constraint is active during growth on these sugars.

**Inhibition of translation highlights the role of environmental signalling in coordination of metabolism in yeast.** Finally, we varied growth rate by translation inhibition by cyclohexamide under controlled glucose batch conditions, and again measured fluxes, growth rate and proteome profiles (Fig. 5a). Upon inhibition of translation, we found a decrease in growth rate and close to proportional decreases in glucose, ethanol and $CO_2$ fluxes, for

both the model and the experimental data (Fig. 5b). Such behaviour is expected when one dominant constraint is active and its extent is varied (cf. glucose-limited fully respiratory growth, Fig. 2). In the case of glucose excess, the model suggested that the cytoplasmic volume was fully occupied with active proteins (minimal UP constraint was hit), and inhibition of translation required higher expression levels of ribosomes, taking away limited proteome space for growth-supporting activities.

However, experimental observations compromised this initial explanation. First, for oxygen the model also predicted a proportional increase with growth rate, but experimentally the fluxes did not change much as did the expression of enzymes involved in oxygen consumption, such as TCA cycle and oxidative phosphorylation (Fig. 5d). Moreover, the ribosomal proteome fraction increased much less with inhibition than the model predicted (Fig. 5c). Since translation inhibition in the model has the same effect as overexpression of a non-functional protein (Supplementary Fig. 7), we followed the earlier observation that the inactive fraction of ribosomes could be recruited for translation, depending on the translational load[16], with only a small improvement (Supplementary Fig. 10).

This suggested that either some constraint prevents the ribosomal fraction from increasing to the optimal levels predicted by the model, or the expression of ribosomes in yeast is dominantly regulated by environmental nutrient signalling and less by internal cues. A dominant role of signalling in ribosomal biogenesis has been suggested before[16]. In yeast the TOR pathway appears to be the master regulator of ribosomal biosynthesis and assembly at steady-state growth[39,40]. Following the TOR-specific targets described by Kunkel[40], we find that key target proteins of this signalling pathway, including ribosomal auxiliary factors, had constant expression levels (Supplementary Fig. 11 and Supplementary Data 6), supporting the dominant role of external rather than internal cues.

When we constrained ribosomal expression to the measured maximal response, ribosomal expression rapidly became the only active constraint in the model, and the proteome space that became available in the cytosol at the lower growth rates was used for increased respiration (Supplementary Fig. 12). This is not observed experimentally, and our data suggest that respiration does not respond to internal cues either. In contrast, the fluxes and expression of proteins involved in glycolysis and amino acid metabolism did decrease with growth rate (Fig. 5b, d). This suggests that these pathways must be sensitive to internal feedback regulation, as is well known for amino acid metabolism[41]. Thus, the proportional fluxes we found for ethanol and glucose upon translation inhibition, are likely the result of control by demand[42], with lower demand at lower growth rate.

## Discussion
In this work, we developed the comprehensive model of a growing, compartmentalised, eukaryal cell to date. It includes all its known metabolic reactions, and details of the protein synthesis, degradation and transport machinery to express the enzymes. The key of our approach is the application of constraints on protein pools in the different compartments that have direct biochemical meaning and could be independently estimated from literature data. Our modelling approach allows reaching a unique level of detail in dealing with cellular compartmentation, in particular of the mitochondria. We furthermore generated a unique set of high-quality quantitative data on both fluxes and the proteome under different, well-controlled, conditions. Through integration and comparison with the model, we provide deeper insight into the physiology of *Saccharomyces cerevisiae*.

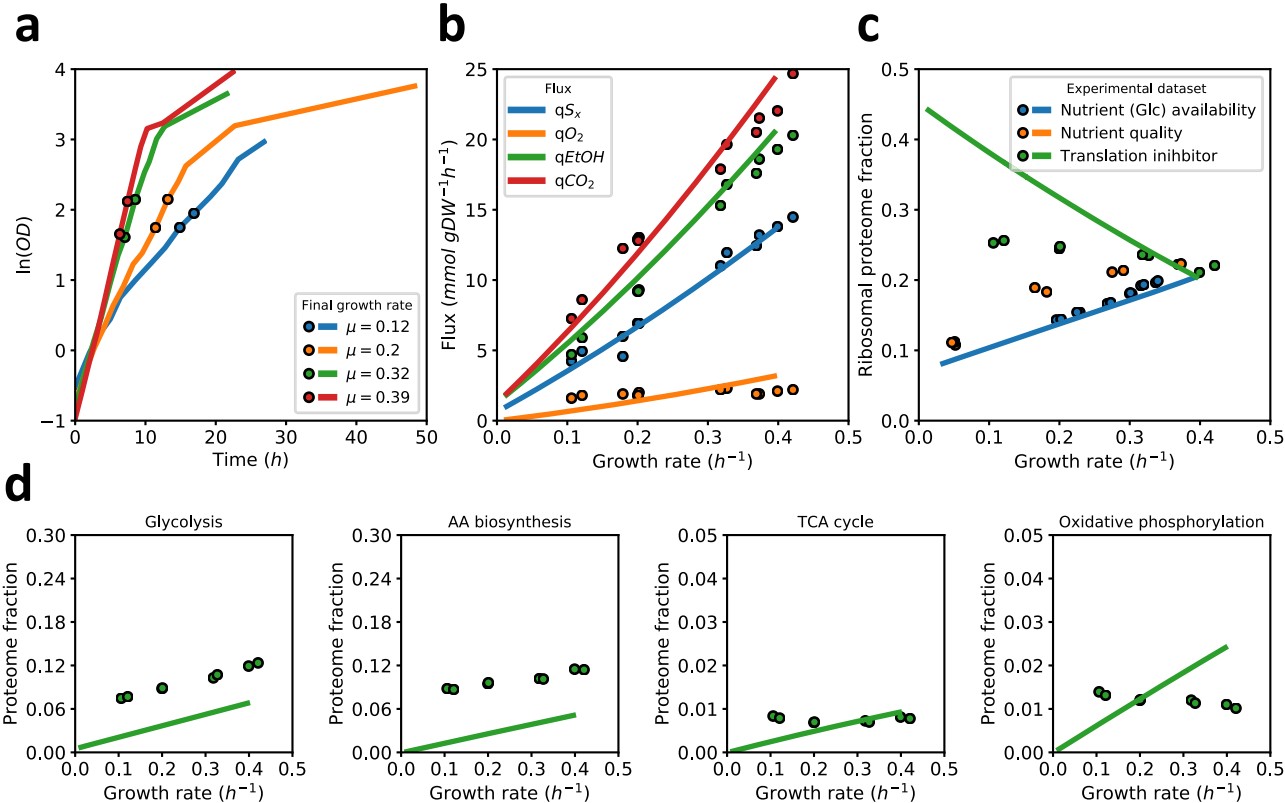

**Fig. 5 The effect of translation inhibition by cyclohexamide on growth rate, fluxes and proteome fractions in controlled aerobic batch fermentations on glucose. a** Dependency of culture optical density (OD) on the time post-inoculation to the medium supplemented with cycloheximide. Lines are values of consecutive OD measurements, points represent the times when cultures were sampled. **b**–**d** Comparison of pcYeast predictions and experimental data: lines are model predictions; symbols are experimental data points. **b** Main catabolic fluxes as a function of the growth rate. **c** Ribosomal proteome fractions. Data from Fig. 1c are included for comparison. **d** Proteome fractions measured for key metabolic pathways, and the minimal proteome fractions predicted by pcYeast. Individual proteins in panel **d** were mapped to metabolic pathways using a manually-curated pathway annotation file (Supplementary Data 5). Source data are provided as a Source Data file.

First, we firmly established that metabolic growth strategies of yeast on glucose can be well understood from a proteome-constrained optimisation problem with growth rate as objective. Through our high resolution sampling around the critical dilution rate, we observed distinct changes in proteins exactly at the onset of ethanol formation in the glucose-limited chemostat. We also show that the active constraints that drive these changes can be different under different conditions such as batch growth on galactose - even if ethanol is made in both cases. Our approach to identify the active cellular constraints may resolve some of the discussion in current literature about *the* cause of overflow metabolism, not only in yeast but possibly also in other eukaryotes, including discussion about the Warburg effect in mammalian cells[43].

Second, the proteome constraints of the model are currently based on experimental observations, but further research could drill deeper into their origin. For example, why would the protein density in the cytosol be relatively constant? Does this balance diffusion rates with catalytic capacities[44]? Are the current morphological dimensions of a yeast cell optimal for growth rate? Recent work on selection for cell number showed that smaller cells can be readily selected for[45]. We also identified that the levels of glucose transport and that of mitochondria need to be constrained to describe the data. Why would yeast not express these components at higher levels? In the case of mitochondria, the proteomics data suggest that rather than a maximum mitochondrial membrane area and matrix volume, there is a maximal rate of mitochondrial biogenesis. Can we calculate this rate from

first principles? One could imagine that an upper limit for mitochondrial growth rate exists if all but eight metabolic proteins need to be transported over the same membrane that must also harbour the full machinery for oxidative phosphorylation. Moreover, we focused on mitochondrial protein content, and ignored details on morphology, lipid synthesis, or possible assembly costs. Thus, a next version of the model will need to address the mitochondrial transport, biosynthesis and morphology in much more detail.

In the case of glucose transport, the model suggested that further increase in glucose transporters beyond wild type expression did not increase growth rate substantially and would likely be invisible for evolution. At maximal saturation of the transporter, glucose transport expression was (just) no longer an active constraint in our model (Fig. 2e). Thus, it appears as if yeast expresses just enough glucose transporters to maximise its growth rate under glucose excess – as found in bacteria[46]. Expressing higher transport levels at lower glucose levels would enhance growth rate but may not pay off if this state is a transient towards glucose starvation, or could be outright dangerous if suddenly glucose would become available[36]. The expression level of the hexose transporters may thus have evolved to be an adaptation to dynamic environments[47]. Long-term evolution experiments in glucose-limited chemostats indeed show gene duplications of high-affinity glucose transporters[48], showing that growth limitation, and hence selection pressure, is on glucose transport under these conditions.

Third, in the case of nutrient uptake limitation, there appears to be excess proteome space that could be filled with anticipatory proteins or heterologous enzymes at no cost in fitness. Even though the composition of such excess proteome space cannot be predicted with our model, we were able to predict metabolic fluxes very well: in this nutrient-limited regime metabolic efficiency (ATP per glucose), not proteome efficiency (ATP per protein), determines the best growth rate strategy. This explains why Flux Balance Analysis applied to only the metabolic network has been so successful, but only under nutrient-limited conditions.

Finally, we found linear or even proportional relationships between growth rate and flux, and between flux and enzyme levels in a sugar excess (batch culture) regime. In terms of regulation analysis[49], such a regime is characterised by hierarchical regulation with absence of metabolic regulation, that is, all changes in flux are brought about by changes in enzyme levels, not their degree of saturation. For glycolysis and amino acid metabolism, the average saturation, estimated as the ratio of the predicted minimal enzyme level to the expressed enzyme level, at maximal growth rate is around 0.5, incidently the level predicted as theoretical optimum for specific reaction rate[50]. In contrast, when growth is limited by glucose availability, the degree of saturation varies and the model suggests a mixture of hierarchical and metabolic regulation, as previously observed in chemostats as well[51].

To conclude, we present a mechanistic, compartmentalised, model of an eukaryal organism in full detail, which can act as a valuable, computable, knowledge base. We show how it can be used to compute protein costs and identify active growth-limiting constraints, and how it can be combined with quantitative flux and proteomics data to provide unprecedented insight into cellular physiology. Finally, we show that also in eukaryal cells, metabolic strategies can be understood on the basis of growth rate optimisation under nutrient and proteome constraints. What remains to be understood is how the cell's signalling and regulatory networks manage to implement these (optimal) proteome allocation strategies.

## Methods

**Strains and shake flask cultivation.** The strain used for this study was *Saccharomyces cerevisiae* strain CEN.PK 113-7D[52]. The stocks used for the experiments were grown in 500 mL shake flask containing 100 mL of YPD medium (10 g L$^{-1}$ of Bacto yeast extract, 20 g L$^{-1}$ of peptone and 20 g L$^{-1}$ of D-glucose). The culture was grown up to early stationary phase and 1 mL aliquots were stored in 20% (v/v) of glycerol at −80 °C. For chemostats, pre-cultures were grown in 500 mL shake flasks containing 100 mL of synthetic medium, the pH was set to 6.0 with 2 M KOH and the medium was supplemented with 20 g L$^{-1}$ of D-glucose[53]. Shake flasks with medium were inoculated with the 1 mL frozen stocks of the strain and the cultivations were performed in an orbital shaker at 200 rpm at 30 °C. Pre-cultures for batches with translation inhibitors were performed using a similar approach, whereas for batches with different carbon sources the pre-cultures were made with the respective carbon sources instead of D-glucose.

**Chemostat cultivations.** Chemostat cultivations were performed in 2 L bioreactors (Applikon, Schiedam, The Netherlands) with a working volume of 1.0 L, the dilution rates used in this study were 0.2, 0.23, 0.27, 0.3, 0.32 and 0.34 h$^{-1}$ in two independent replicate cultures. Growth rates were controlled by modifying the inflow rate on each experiment. Synthetic medium according to Verduyn[53] supplemented with 7.5 g L$^{-1}$ of glucose and 0.25 g L$^{-1}$ Pluronic 6100 PE antifoaming agent was supplied to the bioreactor from a 20 L continuously mixed reservoir vessel. Cultures were sparged with dried air at a flow rate of 700 mL min$^{-1}$ and stirred at 800 rpm. The pH of the cultures was maintained at 5.0 by automatic addition of 2 M KOH. If, after at least six volume changes, the cultures dry cell weight concentration and carbon dioxide production rate differed less than 2% over two consecutive volume changes the cultures were considered to be in steady state. For cultures with dilution rates of 0.27, 0.3, 0.32 and 0.34 h$^{-1}$, cultures were first maintained at a dilution rate of 0.2 h$^{-1}$ for 15 h (3 volume changes) prior to increasing the specific dilution rate to said values.

**Batch cultivations with different carbon sources.** Batch cultivations (two independent replicate cultures) were performed using synthetic medium[53], the medium was supplemented with 20 g L$^{-1}$ final concentrations of the carbon sources, either D-trehalose, D-galactose, D-maltose or D-glucose (Sigma Aldrich). The bioreactors were inoculated with 100 mL of yeast shake flask cultures, exponentially growing on the

specific carbon source. The final OD$_{660}$ of all pre-cultures was 4. Cultivations were performed at 30 °C, the pH was kept at 5.0 by automatic addition of 2 M KOH. The working volume of the bioreactors was 1.4 L in 2 L bioreactors (Applikon, Schiedam, The Netherlands). The cultures were stirred at 8000 rpm and sparged with a flow rate of 700 mL min$^{-1}$ of dried air. Oxygen levels were kept above 40% of the initial saturation level as measured with Clark electrode (Mettler Toledo, Greifensee, Switzerland).

**Batch cultivations with the translation inhibitor cycloheximide.** Batch cultivations (two independent replicate cultures) with the translation inhibitor cycloheximide were performed as for the batches with different carbon sources, except that all the batch cultures ran on 20 g L$^{-1}$ of D-glucose and were supplemented with different concentrations of cycloheximide with the aim of reaching specific growth rates. In total five growth rates were studied, being 0.06, 0.12, 0.2, 0.32 and 0.41 h$^{-1}$ (adding respective cycloheximide concentrations of 228.96, 124.51, 52.15, 25.99 and 0 µg L$^{-1}$).

**Analytical methods.** Cultures dry weight was measured by filtering 20 mL of culture, the sample was filtered in pre-dried and pre-weight membrane filters with a pore size of 0.45 µm (Gelman Science), the filter was washed with demineralised water, subsequently, it was dried in a microwave (20 min, 350 W) and the final weight was measured, the difference being the dry weight in the culture sample.

For the measurement of organic acids and residual carbon source concentrations, supernatants of the cultures were used. For carbon-limited chemostat cultures, the samples were directly quenched with cold steel beads and filtered[54], whereas samples from batch cultures were centrifuged (5 min at 16.000× $g$). The supernatants were analysed by high-performance chromatography analysis on an Agilent 1100 HPLC (Agilent Technologies) equipped with an Aminex HPX-87H ion-exchange column (BioRad, Veenendaal, The Netherlands), operated with 5 mM H$_2$SO$_4$ as the mobile phase at a flow rate of 0.6 mL min$^{-1}$ and at 60 °C. Detection was according to a dual-wavelength absorbance detector (Agilent G1314A) and a refractive-index detector (Agilent G1362A).

The exhaust gas from batch cultures was cooled down with a condenser (2 °C) and dried with a PermaPure Dryer (model MD 110-8P-4; Inacom Instruments, Veenendaal, the Netherlands) before online analysis of carbon dioxide and oxygen with a Rosemount NGA 2000 Analyser (Baar, Switzerland).

**Glycogen and trehalose assays.** 1 mL of culture was taken from the chemostats and directly added to 5 mL of cold methanol (−40 °C). The sample was mixed and centrifuged (4400× $g$, −20 °C for 5 min), the supernatant was discarded, and the pellet was washed in 5 mL of cold methanol (−40 °C), and pellets were stored at −80 °C until further processing. Subsequently, the pellets were resuspended in 0.25 M Na$_2$CO$_3$ and trehalose and glycogen were extracted using boiling water or alkaline/acid extraction, respectively[55,56]. D-glucose released from trehalose and glycogen were measured with a D-glucose assay kit (K-GLUC Megazyme), two biological replicates and three technical replicates were analysed per condition.

**RNA determination.** For RNA determination, 1–2 mL of broth was transferred to a filter (pore size of 0.45 µm, Gelman Science), after which the filter was washed with cold TCA 5%. The cells were resuspended in 3 mL of TCA 5% and centrifuged for 15 min at 4 °C at 3000× $g$ The supernatant was removed and the pellet was stored at −20 °C. Finally, samples were processed using the orcinol method[57]. Two biological replicates and three technical replicates were analysed per condition.

**Protein determination.** For the batches with CHX, culture volumes corresponding to 50 mg of DCW were centrifuged, washed twice with cold demineralised sterile water and divided into two aliquots of 5 mL. 2 mL of the aliquot (containing 10 mg DW) was mixed with 1 mL of 3 M NaOH and incubated at 100 °C for 10 min. The final mix was diluted and processed following the copper-sulfate based method[58]. The absorbance of the supernatant was measured at 510 nm, for calibration lyophilized bovine serum albumin (A2153, Sigma Aldrich) was used. Two biological replicates and three technical replicates were analysed per condition.

**Proteomics sample preprocessing.** Aliquots of 20 mL of culture from chemostats and batches with different carbon sources were centrifuged (3000× $g$, 4 °C, 10 min) and washed two times, the final pellet was flash frozen in liquid nitrogen and stored at −80 °C. Two biological replicates and two technical replicates were analysed per condition.

Frozen cell pellets were thawed on ice before transfer to Precellys® Lysing Kit 2 ml screw cap vials with 0.5 mm glass beads (Bertin Instruments, France). Lysis was performed in 250 µl lysis buffer, 50 mM ammonium bicarbonate with cOmplete protease inhibitor cocktail (ROCHE, Switzerland), using a Minilys Personal Tissue Homogenizer (Bertin Instruments, France), at maximum speed for 15 cycles of 30 s with a 1 min rest on ice between each cycle.

Lysed material was centrifuged for 10 min 13,000× $g$ at 4 °C, the supernatant fraction was removed and retained. Fresh lysis buffer (250 µl) was added to the insoluble material, which was resuspended before extraction from the vial via a small hole inserted into the vial base. Soluble and insoluble fractions were recombined and the total final volume recorded. Protein concentration was determined using Pierce$^{TM}$ Coomassie Plus Bradford Assay Kit (ThermoFisher Scientific, UK).

Protein (100 µg) from each sample was treated with 0.05% (w/v) RapiGest[TM] SF surfactant (Waters, UK) at 80 °C for 10 min, reduced with 4 mM dithiothreitol (Melford Laboratories Ltd., UK) at 60 °C for 10 min and subsequently alkylated with 14 mM iodoacetamide (SIGMA, UK) at room temperature for 30 min. Proteins were digested with 2 µg Trypsin Gold, Mass Spectrometry Grade (Promega, US) at 37 °C for 4 h before a top-up of a further 2 µg trypsin and incubation at 37 °C overnight. Digests were acidified by addition of trifluoroacetic acid (Greyhound Chromatography and Allied Chemicals, UK) to a final concentration of 0.5% (v/v) and incubated at 37 °C for 45 min before centrifugation at $13,000 \times g$ (4 °C) to remove insoluble non-peptidic material.

**Proteomics analytics**. The sample running order was randomised using a random number generator (Random.org).

Samples were analysed using an UltiMate[TM] 3000 RSLCnano system (ThermoFisher Scientific) coupled to a Q Exactive™ HF Hybrid Quadrupole-Orbitrap™ Mass Spectrometer. Protein digests (1 ug of each) were loaded onto a trapping column (Acclaim PepMap 100 C18, 75 µm x 2 cm, 3 µm packing material, 100 Å) using 0.1% (v/v) trifluoroacetic acid, 2% (v/v) acetonitrile in water at a flow rate of 12 µL min-1 for 7 min.

The peptides were eluted onto the analytical column (EASY-Spray PepMap RSLC C18, 75 µm x 50 cm, 2 µm packing material, 100 Å) at 40 °C using a linear gradient of 120 min shallow gradient rising from 8% (v/v) acetonitrile/0.1% (v/v) formic acid (Fisher Scientific, UK) to 30% (v/v) acetonitrile/0.1% (v/v) formic acid at a flow rate of 300 nL min$^{-1}$. The column was then washed at 1% A: 99% B for 8 min, and re-equilibrated to starting conditions. The nano-liquid chromatograph was operated under the control of Dionex Chromatography MS Link 2.14.

The nano-electrospray ionisation source was operated in positive polarity under the control of QExactive HF Tune (version 2.5.0.2042), with a spray voltage of 2.1 kV and a capillary temperature of 250 °C. The mass spectrometer was operated in data-dependent acquisition mode. Full MS survey scans between m/z 300–2000 were acquired at a mass resolution of 60,000 (full width at half maximum at m/z 200). For MS, the automatic gain control target was set to $3e^6$, and the maximum injection time was 100 ms. The 16 most intense precursor ions with charge states of 2–5 were selected for MS/MS with an isolation window of 2 m/z units. Product ion spectra were recorded between m/z 200–2000 at a mass resolution of 30,000 (full width at half maximum at m/z 200). For MS/MS, the automatic gain control target was set to $1e^5$, and the maximum injection time was 45 ms. Higher-energy collisional dissociation was performed to fragment the selected precursor ions using a normalised collision energy of 30%. Dynamic exclusion was set to 30 s.

**Proteomics data analysis**. The resulting raw data files generated by XCalibur (version 3.1) were processed using MaxQuant software (version 1.6.0.16)[59]. The search parameters were set as follows: label free experiment with default settings; cleaving enzyme trypsin with two missed cleavages; Orbitrap instrument with default parameters; variable modifications: oxidation (M) and Acetyl (protein N-term); first search as default; in global parameters, the software was directed to the FASTA file; for advanced identification 'Match between runs' was checked; for protein quantification we only used unique, unmodified peptides. All other Max-Quant settings were kept as default. The false discovery rate (FDR) for accepted peptide spectrum matches and protein matches was set to 1%. The CEN.PK113-7D Yeast FASTA file was downloaded from the *Saccharomyces* Genome Database (SGD) (https://downloads.yeastgenome.org/sequence/strains/CEN.PK/CEN.PK113-7D/CEN.PK113-7D_Delft_2012_AEHG00000000/).

The resulting MaxQuant output was then analysed using the MSstats package (version 3.5.6)[60] in the R environment (version 3.3.3) to obtain differential expression fold changes with associated *p* values, along with normalised LFQ and intensity values[61]. The mass spectrometry proteomics data have been deposited to the ProteomeXchange Consortium via the PRIDE[62] partner repository with the dataset identifier PXD030003.

**Reporting summary**. Further information on research design is available in the Nature Research Reporting Summary linked to this article.

## Data availability

Physiological measurements (specific consumption and secretion rates and yields) are provided in the Supplementary Data 1. Processed label-free quantitative proteomics data of the chemostat and bioreaction cultivations are provided in Supplementary Data 2 to 4. Raw mass spectrometry data are available at the PRIDE database with identifier PXD030003. Source data are provided with this paper.

## Code availability

The full description of the pcYeast model is provided as Supplementary Notes 1–6. The model implementation in Python, together with the code and data used to produce figures of this manuscript (based on the Python implementation) are published on Zenodo[63] [https://zenodo.org/record/5732995]. Alternative model implementation in MATLAB is available on GitHub, SysBioChalmers/Yeast-ME-GEM [https://github.com/SysBioChalmers/Yeast-ME-GEM].

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

## Acknowledgements

We thank Eunice van Pelt-KleinJan, Daan de Groot, and other members of the Systems Biology Lab at the Vrije Universiteit Amsterdam for fruitful discussions on this work, and Michael Nelson for the help with analysing MS-based proteomics data. This work was supported by NWO (NWO ERA-IB-2, project No 053.80.722 to BT and PDL), ERA-IB 4207-00002B DSForsk to JN, and Biotechnology and Biological Sciences Research Council (BB/M025748/1 to SH, BB/M025756/1 to RB). PG and BT also acknowledge support by Marie Skłodowska-Curie Actions ITN "SynCrop" (grant agreement No 764591). We thank SURFsara for the HPC resources through access to the Lisa Compute Cluster.

## Author contributions

Conceptualization, funding acquisition and supervision: B.T., J.N., P.D.L., R.B., S.H.; experimental data collection: A.R.P., V.H., S.W.H.; experimental data analysis: A.R.P., P.G., M.G.A., M.M.M.B.; computational modeling: I.E.E., P.G.; formal analysis: I.E.E., P.G., J.vH., F.J.B., N.S., J.N., P.D.L., B.T.; writing – original draft: B.T.; writing – editing: I.E.E., P.G., F.J.B., J.N., B.T. All authors have read and approved the manuscript.

## Competing interests

The authors declare no competing interests.
