## [Peer Review File · Nature Communications]

Whole-cell modeling in yeast predicts compartment-specific proteome constraints that drive metabolic strategiesReviewers' Comments:

Reviewer #1:

Remarks to the Author:

This manuscript looks at metabolic strategies and phenomena (such as the Crabtree effect) through the lens of whole-cell metabolic model in yeast. The said model considers a comprehensive network of metabolic reactions and tries to optimize growth rate by constraining compartment-specific protein pools (of the cell).

The authors calibrate their model parameters using a number of experiments and previous literature. The work also then compares the predictions from their model to experimental results curated by them.

The authors declare their work as the most comprehensive model of the yeast cell metabolism, and as per my knowledge (I must state that I do not work on yeast) and my reading of their model (Supplementary material), their claim seems robust. They considered the full known metabolic reaction network along with protein synthesis and degradation, in yeast (although the model can be still be made more detailed, such as modeling the mitochondrial processes which is missing in the current form - which they themselves mention at some point). Therefore, their work would be of importance to the field of yeast metabolism, as well as people working on Flux Balance analysis models in general. The mode of their model is not novel, but the amount of data and constraints they put into it makes it noteworthy.

Although I found the paper very interesting and scientifically sound, it was poorly written in parts and sometimes, repetitive. The overall structuring of the narrative in terms of what is known, why it was necessary and what they did (and why it was special) made sense intuitively, and the flow can still be improved to make the reader orient themselves more. One big plus point of the paper was that the text already listed the assumptions and shortcomings, and therefore, it was easy to see what parameter values were incorporated or constrained by hand and what parameters were explored. All that the paper, according to me, lacks now is a more tight writing - especially of the introduction. (on a smaller note - many punctuations are misplaced, and compound statements are unnecessarily introduced throughout the work).

The conclusions drawn based on the models and the experimental data were sound, to the best of my knowledge. The main features of a such an optimization model comes from constraints placed upon them, and the authors do a good job of describing and implementing them (based on the description in supplementary notes, as I cannot see the code. However, I feel that one can replicate the model from the description). The experiments also are well described and seem reproducible.

Reviewer #2:

Remarks to the Author:

The authors present a next generation metabolic reconstruction of *Saccharomyces cerevisiae* by including a detailed description of the protein synthesis machinery based on cellular compartment. I found the manuscript to be well written and very insightful, especially with respect the modeled Crabtree effect. Below I have included a few comments meant to improve the clarity of the presentation.

1. Most bacterial metabolic constructions focus on a particular strain. Here the authors develop a generic *S. cerevisiae* model that is strain independent even through carbon utilization in *S. cerevisiae* can be strain dependent. I believe this point is worth mentioning.
2. I believe this comment requires a brief clarification: "We use a binary search algorithm to find the

maximum growth rate where the linear programming problem becomes just infeasible". Why would the LP become "just" infeasible?

3. Can the authors comment on the modeling of the biomass composition and whether the composition is changed from previous studies due to expansion of the model?

4. Figure 4d: The authors may want to consider somehow denoting the individual enzymes shown.

Reviewer #3:

Remarks to the Author:

Review of Elseman, Rodrigues Prado, Grigaitis et al

The authors present a whole-cell model of yeast to understand how compartment specific constraints of how the proteome influences metabolism of yeast. The authors put in a lot of effort, incorporating individual compartments and make several new measurements attempting to quantify their assumptions. Overall, however, there are major issues in both scientific content and impact, and I unfortunately do not think the paper is suitable for publication at this stage. Even if the technical and scientific issues were resolved, I am doubtful that this work constitutes a significant enough advance to justify publication in Nature communications.

Major issues:

Most worrisome, there is a pattern throughout the paper of overstating the conclusions that can be drawn from their approach, data and the analysis. Rather than proving that certain constraints are at work, the work simply presents data that is consistent with a presented model based on these assumptions. If published, the manuscript and the abstract should certainly be reformulated to make this clear.

A major scientific question is whether this work actually 'predicts' compartment-specific proteome constraints. I see several problems here. First, the authors assume "all metabolically-active proteins are modeled to work at their maximal rate and are minimally expressed." This constraint is typically used, but it does not match reality. It is known that biosynthetic processes work below their maximum efficiency and even recognized by the authors themselves (see Fig. 3, but also works of many other, e.g. on ribosomes). Because the authors claim is that they can predict when proteome fractions hit certain constraints, the authors need to explain clearly why they believe that this implementation of the model has predictive power. Most proteome fractions are clearly wrongly predicted (Fig. 3, Fig. S4, etc), so how can the model correctly predict when these fractions hit a constraint?

Second, how do the authors 'predict' the effect of growth rate gratuitous protein expression? What is the interpretation of the X-axis intercept? How does this depend on the authors' choice of the unspecified protein fraction ($UP = 0.25$) and other parameters that are a crucial for the model? Defining parameters and laying out their calculation transparently is important, because it allows the reader to judge if this is a true 'prediction'. As is, UP seems to be hand-picked.

Third, a central prediction of the authors is that metabolism hits certain constraints, such as 'unspecified protein' or 'mitochondrial membrane'. How do the authors know where these constraints are, and whether they are really uncircumventable? The back of the envelope calculation that the authors make for the maximal mitochondrial is highly questionable. Based on the mitochondrial membrane surface area of yeast growing on YPD, $15\mu\text{m}^2$, times the 6-fold increase between glucose and Ethanol medium, times the ratio of protein mass to lipid mass, 80%, the authors estimate the maximum mitochondrial membrane must be $72\mu\text{m}^2$, and mitochondrial membrane becomes limiting at a growth rate of 0.28/h. This calculation is highly questionable.

The ratio of protein mass to lipid mass is not a packing ratio. Why does this calculation involve a measurement on YPD? And how do the authors know that this is the 'maximum'? In fact, yeast growing on Ethanol medium can make 6x more mitochondrial membrane than on glucose. Therefore, it seems that mitochondria are not limiting during glucose growth. Because glucose steady-state growth is the final condition chosen, a much better estimate for the maximum mitochondrial surface area would be 6 times that required of the final steady state in the model. Because, according to the model mitochondrial membrane is 50% below its constraint (Fig. 2e), we would thus expect that in the intermediate regime (blue) mitochondria are not limiting.

Either way, I see no evidence that the authors actually know what the maximum mitochondrial membrane area is (point 'Third') and nor that they exactly predict the proteome fractions (point 'First'). Thus, the authors are unable to predict which constraint becomes limiting when.

Unfortunately, I must conclude that the manuscript does not support the claims of the authors. Even if the authors were to somehow 'improve' the constraints, I see no possibility that they can predict when proteome constraints become limiting. Rather, the model recapitulates certain flux phenotypes based on assumptions on proteome constraints. This does not seem insufficient for a journal like Nature Communications to me. I encourage the authors to remove the 'prediction' aspect of the paper and to revisit their constraints. If the constraints are not well-defined, they can be used as fitting parameters. The resulting paper will still be of interest to the community, and I hope to see it published in a different journal.

Minor issues:

Throughout the paper, the presentation of math can be improved. I encourage the authors to carefully implement standard typography practices. For example, variables in italic, vectors in bold, units in roman font type. 'Dots' and 'crosses' should be used with caution, as they can be misinterpreted as cross products and dot products. Here is a reference for the IUPAC standard: <https://iupac.org/cms/wp-content/uploads/2016/01/ICTNS-On-the-use-of-italic-and-roman-fonts-for-symbols-in-scientific-text.pdf>

In addition, the authors should define every variable where it is used (either in main text or figure caption). At several instances it is not clear what the mathematical formulation means unless one reads the SI (e.g. $S_{xv} = 0$ in Fig. 1b).

At several instances in the paper the range of the plot is too large, so that data cannot be assessed. E.g. Fig. 1C&D, the authors should show the regime of data, i.e. Ribosome fraction from 0 to 0.3 or proteome mass fraction from 0 to 0.2. In Fig. 2A there is a similar issue. It is impossible to judge if the model is a good or a bad fit.

In Fig. 4d, the authors make the statement that proteome fractions change linearly, but the reader can only judge the most abundant proteins. This data should be presented better. One option could be normalizing data.

"The maximal feasible growth rate that the model predicted can be linked directly to the dilution rate in the chemostat, allowing comparison of model prediction and data (Fig. 2a). "

This panel is missing. Also, it would be great to know what this sentence is referring to. Does the model growth rate in a chemostat match the dilution rate?

"The (residual) glucose concentrations were calculated from documented (high) affinity of the

transporters, which is close to 1 mM"

I did not understand this sentence. Please explain how that was done.

The resulting relationship between growth rate and residual glucose concentration fit experimental data very well (Fig. 2b), validating our expectation that we could ignore intracellular glucose³²

In batch growth, where there is excess glucose, wouldn't we expect that growth rate is independent of glucose concentration? Does this only refer to the chemostat? Also, this seems to refer to Fig. 2A. How about plotting $\log(\text{glucose concentration})$ versus growth rate? This would make the plot resemble a typical enzyme kinetics plot and could be easier to read. And how do the authors conclude that they can ignore internal glucose?

A discussion of Fig. 2B is missing.

In Fig. S1 it is not clear what 'relative' is referring to.

What are the lines in Fig. 5B?

Reviewer #1 (Remarks to the Author):

This manuscript looks at metabolic strategies and phenomena (such as the Crabtree effect) through the lens of whole-cell metabolic model in yeast. The said model considers a comprehensive network of metabolic reactions and tries to optimize growth rate by constraining compartment-specific protein pools (of the cell).

The authors calibrate their model parameters using a number of experiments and previous literature. The work also then compares the predictions from their model to experimental results curated by them.

The authors declare their work as the most comprehensive model of the yeast cell metabolism, and as per my knowledge (I must state that I do not work on yeast) and my reading of their model (Supplementary material), their claim seems robust. They considered the full known metabolic reaction network along with protein synthesis and degradation, in yeast (although the model can be still be made more detailed, such as modeling the mitochondrial processes which is missing in the current form - which they themselves mention at some point). Therefore, their work would be of importance to the field of yeast metabolism, as well as people working on Flux Balance analysis models in general. The mode of their model is not novel, but the amount of data and constraints they put into it makes it noteworthy.

We appreciate the positive Reviewer's impression on our manuscript.

Although I found the paper very interesting and scientifically sound, it was poorly written in parts and sometimes, repetitive. The overall structuring of the narrative in terms of what is known, why it was necessary and what they did (and why it was special) made sense intuitively, and the flow can still be improved to make the reader orient themselves more. One big plus point of the paper was that the text already listed the assumptions and shortcomings, and therefore, it was easy to see what parameter values were incorporated or constrained by hand and what parameters were explored. All that the paper, according to me, lacks now is a more tight writing - especially of the introduction. (on a smaller note - many punctuations are misplaced, and compound statements are unnecessarily introduced throughout the work).

Thank you for the attention to details and your suggestion. We have checked and improved punctuations and tightened parts of the manuscript to enhance readability and precision of our statements. Since the remark on poor writing was rather general, we hope to be able to work with the editor to meet the journal's standards.

The conclusions drawn based on the models and the experimental data were sound, to the best of my knowledge. The main features of a such an optimization model comes from constraints placed upon them, and the authors do a good job of describing and implementing them (based on the description in supplementary notes, as I cannot see the code. However, I feel that one can replicate the model from the description). The experiments also are well described and seem reproducible.

We thank the Reviewer for positively evaluating the soundness of our findings and modeling approach.

Reviewer #2 (Remarks to the Author):

The authors present a next generation metabolic reconstruction of *Saccharomyces cerevisiae* by including a detailed description of the protein synthesis machinery based on cellular compartment. I found the manuscript to be well written and very insightful, especially with respect the modeled Crabtree effect. Below I have included a few comments meant to improve the clarity of the presentation.

Thank you for the positive evaluation of the manuscript.

1. Most bacterial metabolic constructions focus on a particular strain. Here the authors develop a generic *S. cerevisiae* model that is strain independent even through carbon utilization in *S. cerevisiae* can be strain dependent. I believe this point is worth mentioning.

We thank the Reviewer for raising a suggestion on an issue that we were indeed not explicit about. Indeed, the yeast metabolic network is strain-independent, and remains so after introducing the 'proteome-constrained'

part. Meanwhile, we used a specific strain, CEN.PK 113-7D, in our experiments and used strain-specific experimental data to model the biomass composition of CEN.PK 113-7D cells in glucose-limited chemostats (Supplementary Notes, SN Table 1). Indeed, strain specificity arises from specific cellular make-up and regulation - reflected in different constraints that can become active at different conditions.

We included a note in the manuscript on this, at the end of the section "Construction of a comprehensive proteome-constrained yeast model" (L147-9):

"<...>the model returns all the flux values associated with the maximal feasible growth rate. **It should be noted that the structure of the pcYeast model is strain-independent: this allows subsequent calibration of the model to accommodate and account for differences in cell physiology and metabolism, inherent to any specific strain of *S. cerevisiae*.**"

In the following section, we explicitly wrote which strain was used for experiments (L152-3):

"We performed a series of experiments, **using a wild-type *S. cerevisiae* strain CEN.PK 113-7D**, for collection of high-quality datasets of fluxes and protein levels, used either as model input or for comparison with model predictions."

2. I believe this comment requires a brief clarification: "We use a binary search algorithm to find the maximum growth rate where the linear programming problem becomes just infeasible". Why would the LP become "just" infeasible?

We apologize for the confusing phrasing; here, "just" was supposed to mean that we obtain the prediction of the maximal growth rate that still results in a feasible LP, but a marginal increment of the growth rate would result in an infeasible LP. In current model implementation, the difference between the growth rates, corresponding to 'highest feasible' and 'infeasible' LPs, respectively, is set to $\Delta\mu = 10^{-4} h^{-1}$. Given this small difference, we believe that we determine the maximal growth rate with high enough precision.

We have changed this sentence (L143-6) as follows:

"We use a binary search algorithm to find the maximum growth rate where **the linear program is still feasible, and a marginal increase in the growth rate would result in an infeasible** the linear programming problem becomes just infeasible; the model returns all the flux values associated with the maximal feasible growth rate."

We also included the following sentence into the Supplementary Notes, 1st paragraph of the Section 2:

"We consider the simulated growth rate to be the highest feasible (μ_{max}) if a very small increment in growth rate $\Delta\mu$ results in an infeasible linear program (i.e. at $\mu = \mu_{max} + \Delta\mu$). We set $\Delta\mu \leq 10^{-4} h^{-1}$ in our simulations."

3. Can the authors comment on the modeling of the biomass composition and whether the composition is changed from previous studies due to expansion of the model?

In the pcYeast model we used growth rate-dependent biomass measurements of the CEN.PK 113-7D cells in glucose-limited chemostats (Supplementary Notes, Table SN1) to determine the stoichiometric coefficients of the biomass components, instead of using the coefficients from the Yeast7.6 biomass equation. We did so based on the observations of Canelas et al. 2011 (PMID 21354323) that the dry biomass composition of CEN.PK 113-7D strain is highly dependent on the growth rate. Answering the second part of the question, the resulting growth rate-dependent biomass equation was altered in two ways due to explicit modeling of protein turnover processes.

First, in conventional genome-scale models, the demand of proteins in the biomass equation is modeled through dilution-by-growth of amino acids: amino acids are charged onto the respective tRNAs and resulting amino acid-tRNAs are consumed through biomass flux, retrieving free tRNAs as products. In the pcYeast model, both protein degradation and dilution-by-growth are handled explicitly, thus we do not include protein demand into the biomass equation as described above.

Second, with explicit modeling of protein synthesis, folding and degradation, a graspable share of previous cellular maintenance-associated energy consumption (ATP maintenance) is being covered (ca. 40%, as suggested by Lahtvee and colleagues, PMID 28365149). Thus, in the pcYeast model, the requirement of the growth-associated maintenance is modeled as described in the Supplementary Notes, Growth-related Constraint 2 (Page 29 of the Supplementary Materials file).

We have now explicitly referred to this section of the model documentation in the Main Text, Section "Calibrating the model against experimental data", L159-61.

4. Figure 4d: The authors may want to consider somehow denoting the individual enzymes shown.

We thank the Reviewer for this suggestion: together with the point raised by Reviewer #3 (Minor Issue 3), we have changed the Figure 4d to both normalize the data and include gene names of the proteins presented.

Reviewer #3 (Remarks to the Author):

Review of Elsemann, Rodriguez Prado, Grigaitis et al

The authors present a whole-cell model of yeast to understand how compartment specific constraints of how the proteome influences metabolism of yeast. The authors put in a lot of effort, incorporating individual compartments and make several new measurements attempting to quantify their assumptions. Overall, however, there are major issues in both scientific content and impact, and I unfortunately do not think the paper is suitable for publication at this stage. Even if the technical and scientific issues were resolved, I am doubtful that this work constitutes a significant enough advance to justify publication in Nature communications.

Major issues:

Most worrisome, there is a pattern throughout the paper of overstating the conclusions that can be drawn from their approach, data and the analysis. Rather than proving that certain constraints are at work, the work simply presents data that is consistent with a presented model based on these assumptions. If published, the manuscript and the abstract should certainly be reformulated to make this clear.

We are sorry to read this, especially because we took great care in being transparent. Reviewer 1 in fact complimented us with this:

"One big plus point of the paper was that the text already listed the assumptions and shortcomings, and therefore, it was easy to see what parameter values were incorporated or constrained by hand and what parameters were explored."

We believe that there is an important fundamental difference at play here about the expectations of the model we present. We strongly believe in the well-known motto by George Box: all models are wrong, and some are useful. We believe our model is useful (see also Reviewer #2, who wrote *"I found the manuscript to be well written and very insightful, especially with respect the modeled Crabtree effect"*). Here is our general reasoning:

We put together, from literature, all processes related to metabolism and protein synthesis, biomass composition, and catalytic rates, into a mathematical model. We use some experimental data to fit crucial parameters, such as those related to translation. We then run the model, and get predictions. Those we compare with independent data. If they "simply" turn out to be consistent, we have confidence in the model and the underlying assumptions. Now the model can be used (useful) to learn new biology. Central to our story are the potential constraints that limit the growth rate. These are genuine predictions by the model, as we run into those when we optimize. We agree these predictions should be validated, and to the best of our abilities we provide a number of observations that actually do this, in our opinion:

1. Under glucose limited chemostats, glucose transport expression is predicted to be the active constraint. Under long-term chemostat cultivation, indeed mutants with gene duplications in HXT genes are the first to be selected [ref 49].
2. Under galactose growth, we observe that laboratory evolution moves towards the predicted naïve behavior the model predicts.
3. Under glucose batch, where we predict the cytosol to be full, we indeed predict a lower growth rate upon expression of the heterologous mCherry protein.

This does not appear to be the proof that the reviewer has in mind. One option we also considered, is to change parameters that affect these constraints, directly and specifically. The mCherry overexpression is one such example, the effect of which we could predict and explain from a total proteome constraint. The other perturbation we could think of and implemented, was with the translation inhibitor. Such direct perturbation experiments, however, turned out to be extremely difficult due to the signaling capabilities of yeast, as we openly discuss.

We therefore feel we did the best we could within reasonable bounds to show the usefulness of our model, and with reviewer 1 we believe we have been really honest about its limitations, and respectfully disagree that there is a pattern of overstating conclusions. We were very careful in stating our promise at the end of the introduction: "Comparison of the model predictions with the data gives unprecedented insight into our physiological understanding of this important model organism." We do not claim that our predictions are always correct, but they give insight.

A major scientific question is whether this work actually 'predicts' compartment-specific proteome constraints. I see several problems here. First, the authors assume "all metabolically-active proteins are modeled to work at their maximal rate and are minimally expressed." This constraint is typically used, but it does not match reality. It is known that biosynthetic processes work below their maximum efficiency and even recognized by the authors themselves (see Fig. 3, but also works of many other, e.g. on ribosomes). Because the authors claim is that they can predict when proteome fractions hit certain constraints, the authors need to explain clearly why they believe that this implementation of the model has predictive power. Most proteome fractions are clearly wrongly predicted (Fig. 3, Fig. S4, etc.), so how can the model correctly predict when these fractions hit a constraint?

As the Reviewer correctly indicates, the situation where all the metabolically active proteins work at their maximal rate is not possible, and therefore additional modeling decisions were needed to capture the effects of undersaturation. This is a really complicated issue due to the nonlinearity of enzyme kinetics, and all the approaches used so far have serious limitations, as the reviewer rightfully points out. The most widely used approach is to determine the apparent turnover values k_{app} individually for all metabolic proteins, using quantitative proteomics data (see e.g. Chen and Nielsen, 2021 PNAS; PMID 34341111). It assumes a constant level of saturation, ignoring the fact that saturation is a function of conditions, as we clearly show with our data. Moreover, we would then use our data to fit the model, rather than to learn from it.

Therefore, we decided in the paper to compare the minimal demands of protein (by assuming the maximal efficiency) with the experimentally determined protein abundance, and use the difference as a proxy for the level of saturation of pathways. How can we then still determine which fraction limits growth? The answer is the artificial, 'unspecified' protein (UP) with no metabolic function and average amino acid composition. This protein is included in all the appropriate proteome-related constraints, such as total protein volume constraint, to ensure that the protein content of the cell remains high. So at low growth rates, with low metabolic rates (and low minimal enzyme levels in the model), the UP expression is high (Fig 2E, upper left panel, but see also Figure 1 of this rebuttal, which we also included in the revised manuscript). A lot of this UP is in fact metabolic enzymes, as we can see from the experimental data, and therefore the discrepancy between predicted minimal levels and expressed levels is high, and saturation of the enzymes low, as expected. This discrepancy is not a failure of the model!

From this, we for example learned that cytosolic proteins seem more undersaturated than mitochondrial proteins, which we found an interesting finding. It seems that the Reviewer mistakenly took those predicted minimal levels as the true predicted levels in the cell, and we added some remarks in the text to make this point clearer.

Section "Proteomics data validates model predictions", L234-7:

"We subsequently measured protein levels with quantitative proteomics and compared them to the **minimal protein levels that the model predicted to be needed to support metabolic flux. Since we compute minimal levels as if all the enzymes worked at their maximal rate, we model predictions.** We expected to underestimate most proteome fractions, ~~because the model predict minimal protein levels required to support metabolic flux.~~"

L240-3:

"The difference between the predicted minimal level and the data may be interpreted as a proxy for the average saturation of enzymes. **In terms of protein synthesis costs, the discrepancies between the experimentally measured enzyme expression and the predicted minimal expression level are covered by the expression of the UP.**"

Figure 1. Simulation of required metabolic protein fraction and the UP fraction as a function of growth rate to ensure the proper protein content of the cell.

As growth rate increases, and the metabolic fluxes and minimal metabolic enzyme fractions do too, UP will become smaller and smaller, until it hits a preset minimal level (see below for further elaboration on picking the minimal fraction of the UP in the proteome). From the proteome point of view, at this point the cytosol is “full” with maximally active proteins, indicating that the total proteome begins to actively limit growth. Subsequently, the model predicts the respiratory proteins being traded-in for fermentative proteins at $\mu > 0.35 h^{-1}$ in glucose-limited chemostats, which is in line with the proteomics data we collected. This therefore is a genuine prediction of the model that is validated by the proteomics data.

Second, how do the authors ‘predict’ the effect of growth rate gratuitous protein expression?

To titrate the amount of a gratuitous protein, we performed a series of simulations where we add a flux coupling constraint which forces flux through the protein synthesis reaction ‘mCherry_translation_Ribosome’ (consuming charged amino acid-tRNAs and energy equivalents [ATP, GTP]). The right-hand-side value of this constraint corresponds to a certain fraction of total proteome. As for all the rest of simulations, we then we determine the highest feasible growth rate for each expression level of the gratuitous proteins, and then compare the growth rate with the unperturbed state (simulation without gratuitous protein expression).

This is now explained in Supplementary Notes, within the newly included subsection “Proteome Constraint 8: Gratuitous protein expression” and the legend of Fig. S7:

“Fig. S7. Predicted metabolic fluxes for dose-dependent mCherry overexpression in glucose minimal medium (simulation performed analogously to the simulation of mCherry overexpression in SD medium, Fig. 1D, points) and inhibition of translation (lines). The overexpression of mCherry was simulated by setting the desired proteome mass fraction, which mCherry should occupy and running the binary search algorithm to determine the highest growth rate which would still result in a feasible linear program. For inhibition of ribosomes, same procedure was applied, but instead of pre-setting the level of gratuitous protein expression, the activity factor of ribosomes ($0 < f_{Ribosome}^A \leq 1$) was varied (see Ribosome Capacity Constraint 1).”

What is the interpretation of the X-axis intercept? How does this depend on the authors’ choice of the unspecified protein fraction (UP = 0.25) and other parameters that are a crucial for the model?

The intercept of the X-axis would be (hypothetical) gratuitous protein expression level at which growth would no longer be possible. This point is still far away and we do not want to extrapolate too far beyond experimental observations. Answering the second part of this question, varying UP values has little impact on the *relative* decrease of the growth rate by mCherry expression. Naturally, the growth rate at the unperturbed state (no

mCherry expression) will differ in absolute terms depending on the UP chosen. To the best of our understanding, the parameters that could influence the outcome of simulating titration of mCherry expression are those related to the rates of protein synthesis, folding and degradation. The parameter that is likely to have the greatest impact, in our experience, is the peptide elongation rate of the ribosomes.

We added this information to the legend of Fig 1 and Fig S7 (see the updated legend of Fig S7 above) where these simulations are described:

“Fig 1. <...> d. Impact of mCherry protein overexpression on growth rate. Symbols show experimental data²⁶, solid lines show model predictions based on glucose minimal (SD) medium or rich SC/YPD media. **Model predictions were obtained by varying the proteome mass fraction, occupied by mCherry, and determining the maximal predicted growth rate at each value of the mass fraction. The relative growth fitness represents the ratio between the growth rate at certain mCherry expression level vs. the unperturbed state (no mCherry expression).”**

Defining parameters and laying out their calculation transparently is important, because it allows the reader to judge if this is a true ‘prediction’. As is, UP seems to be hand-picked.

We fully agree and did our best to be as transparent as possible. Given the complexity of the model, however, we understand that there might still be some parameters that remain poorly explained, for which apologize. The minimal UP level stands out.

In the current manuscript, the choice of the minimal UP value (the parameter rendered as ‘hand-picked’) is indeed presented as a modeler’s choice, but we would like to emphasize that the parameter value is, in fact, an informed choice and based on the quantitative proteomics data we collected. We realize now that we should have included this valuable piece of information more explicitly into the manuscript.

To better explain the choice of the UP value, we will include Figure 2 (below) in the Supplements, where we plot proteome fractions ($g/g \text{ total protein}$) from experimental data and model predictions.

Figure 2. Computed protein fractions as a function of growth rate for the proteome turnover-associated proteins (blue) and proteins, participating in other cellular processes (orange). The dots represent experimentally determined proteome fractions from glucose-limited chemostats (at $D = 0.20$ to 0.34 h^{-1}) and trehalose- and glucose-excess batch cultivations ($\mu = 0.043$ and $\mu = 0.399 \text{ h}^{-1}$; left-most and right-most points, respectively), and lines are model predictions in glucose-limited conditions. The annotations for the “Proteome turnover” class were collected by the Authors (Table S1 of the Manuscript); for “Metabolic proteins”, we used KEGG ontology annotations for processes related to metabolism.

For this, we attributed proteins to coarse proteome sectors. The first category of proteins was associated with proteome turnover (similar to Terry Hwa's "R" fraction): translation factors, ribosomal proteins, chaperones, proteasome subunits/proteases (in all cases, including not only cytoplasmic, but also their mitochondrial counterparts). The remaining proteins from experimental datasets were attributed to another category ("Rest of cellular processes", a combination of Hwa's "P" and "Q" sectors). In the case of experimental data (points in the Figure 2), these proteins comprised not only metabolic proteins (shown as orange triangles in the plot) but also proteins with other functions (such as signaling and structural proteins).

Then we compared the experimental measurements with model simulations, where the lines show either, again, the proteome fraction of proteins, involved in protein turnover, or - for the "Rest of cellular processes" - the sum of proteome fractions occupied by metabolic proteins and the UP. Notably, with the minimal UP value of $0.25 \text{ g UP/g protein}$, we do see an agreement of the model simulations to the experimental data for both the protein fractions, assigned to both "Proteome turnover" and "Rest of cellular processes" categories. Therefore, we believe that the data we present does in fact support our decision to use the UP value of $0.25 \text{ g UP/g protein}$ for model simulations.

As acknowledged above, we included the Figures 1 and 2 of this rebuttal to the Manuscript as the Supplementary Figure 1 and shifted the position of all the rest of the Supplementary Figures. We included the explanation of the Rebuttal Figure 2 we provided above to the Supplementary Notes, Section "Proteome Constraint 7: Unspecified protein" (not provided here as it is almost 1:1 copy).

Third, a central prediction of the authors is that metabolism hits certain constraints, such as 'unspecified protein' or 'mitochondrial membrane'. How do the authors know where these constraints are, and whether they are really uncircumventable?

Answering the first part of the Reviewer's question, we formulated constraints at the level of protein expression, and all proteins have specific locations in the cell that were based on careful manual annotation of these proteins. Compartment-specific proteome expression constraints (e.g. constraint for carbon transport, mitochondrial membrane area, mitochondrial matrix volume etc.) are weighted sums of these proteins, and upper limit values were determined by consulting literature measurements, or by using back-of-envelope calculations.

These decisions are all documented in the Supplementary Notes with accompanying references to the original studies or indicating that the parameters were chosen by the Authors. As we use an optimization, we will necessarily hit the constraints that bound the optimum, and these are the constraints predicted by the model to be active. For example, the model hit the minimal UP constraint at 0.35 h^{-1} when we gradually increased the growth rate through the saturation of the glucose transporter (Figure 1 of this rebuttal). This then forces the model to change its behavior, and this change is also observed experimentally.

The 'uncircumventability' of these constraints is an important and interesting question, and the general answer should be: it depends. Many constraints are probably circumventable in evolutionary sense: yeast can express more transporter, or could possibly change morphology. But this will probably compromise other abilities. Other constraints, such as the maximal protein density, has a physico-chemical basis that is out of evolutionary reach. Comparison with laboratory evolution experiments is an important way to tell how hard a constraint is, and proteome-constrained models provide a tool to integrate all the knowledge and data to understand these constraints better, as we also showed for *Lactococcus lactis* recently (Chen et al, Mol Syst Biol 2021).

The back of the envelope calculation that the authors make for the maximal mitochondrial is highly questionable. Based on the mitochondrial membrane surface area of yeast growing on YPD, $15 \mu\text{m}^2$, times the 6-fold increase between glucose and Ethanol medium, times the ratio of protein mass to lipid mass, 80%, the authors estimate the maximum mitochondrial membrane must be $72 \mu\text{m}^2$, and mitochondrial membrane becomes limiting at a growth rate of 0.28/h. This calculation is highly questionable.

We agree the calculation is not rock solid, but it is the best we could do. We put as much existing biological knowledge into the model as we could – in order to later validate predictions with newly generated experimental data. Maximal mitochondrial inner membrane surface area is one of the cases where strong experimental evidence and quantitative measurements were lacking, and thus we had to restrain to the presented back of the envelope calculations, which we will try to justify below.

The ratio of protein mass to lipid mass is not a packing ratio. Why does this calculation involve a measurement on YPD? And how the authors know that this is the ‘maximum’?

We tried to get the best estimate of the maximal mitochondrial content that yeast can express, and these YPD measurements were available and the best we have. We then see that using these numbers – on which we are fully transparent – result in a good prediction of flux behavior. Moreover, considering the mitochondrial proteome fractions (Fig. 3B), we do see that the fraction of mitochondrial proteins is the highest at the experimentally determined critical dilution rate; with more rapid growth, the mitochondrial proteome starts to decline. The model, given the input of the mitochondrial inner membrane area maximum of $72 \mu\text{m}^2$, captures this behavior. Therefore, the $72 \mu\text{m}^2$ seems a fair estimate for the parameter. Note that we acknowledge that the mitochondrial parameters require a more detailed inquiry in model updates, and that the data suggests that the rate of mitochondrial biosynthesis may actually be the limiting factor.

In fact, yeast growing on Ethanol medium can make 6x more mitochondrial membrane than on glucose. Therefore, it seems that mitochondria are not limiting during glucose growth.

This comment suggests that the Reviewer unfortunately missed the key point of the paper: that constraints are condition- and compartment-specific. When growing on excess glucose – in batch – mitochondria are repressed: this is observed experimentally, and our model suggests that the reason is to make room for biosynthetic machinery required to grow at the high growth rate under glucose-excess conditions. So indeed, we agree that mitochondria are not limiting under these conditions, the cytosolic space is. We also agree that yeast can make more mitochondria when grown (more slowly) on ethanol than grown on excess glucose. Our data and model suggest that the maximal mitochondrial membrane, set by growth on ethanol, limits glucose growth at a growth rate of 0.28 h^{-1} (but not at 0.4 h^{-1} !). Indeed, based on our mitochondrial proteome fraction (Fig 3B), mitochondrial protein content at 0.28 h^{-1} is about 4 times higher than at 0.4 h^{-1} . We hope that this clears up the confusion!

Because glucose steady-state growth is the final condition chosen, a much better estimate for the maximum mitochondrial surface area would be 6 times that required of the final steady state in the model.

Thank you for the suggestion, but that is exactly how we got to the maximal membrane constraint, so we agree here! We used six times the final steady state, which is growth on YPD glucose. Growth rate is 0.45 h^{-1} on YPD glucose, and 0.4 h^{-1} on minimal media, and if we extrapolate the drop in mitochondrial proteins, then the mitochondrial content on YPD must be lower than on minimal media. Then the 6 times on YPD reported in literature is consistent with our proteomics data that show that maximal mitochondrial protein expression is 4 times higher than on batch glucose minimal media (see also answer above). We agree that the numbers we provide are not rock solid, but very reasonable and consistent with the available data.

We now refer to these calculations explicitly in the Supplementary Notes, “Proteome Constraint 4: Inner mitochondrial membrane constraint”.

Because, according to the model mitochondrial membrane is 50% below its constraint (Fig. 2e), we would thus expect that in the intermediate regime (blue) mitochondria are not limiting.

This is the same argument as for ethanol growth, which we explained is a misunderstanding from the reviewer that we have hopefully resolved.

Either way, I see no evidence that the authors actually know what the maximum mitochondrial membrane area is (point ‘Third’) and nor that they exactly predict the proteome fractions (‘point ‘First’). Thus, the authors are unable to predict which constraint becomes limiting when.

We hope we have been able to explain the usefulness of the modeling approach, and to take away the main concerns and unclarities.

Unfortunately, I must conclude that the manuscript does not support the claims of the authors. Even if the authors were to somehow 'improve' the constraints, I see no possibility that they can predict when proteome constraints becoming limiting.

We hope we have been able to explain that this conclusion was based on partly misunderstanding of the work, and that the reviewer can agree to adjust his or her judgement.

Rather, the model recapitulates certain flux phenotypes based on assumptions on proteome constraints. This does not seem insufficient for a journal like Nature Communications to me. I encourage the authors to remove the 'prediction' aspect of the paper and to revisit their constraints. If the constraints are not well-defined, they can be used as fitting parameters. The resulting paper will still be of interest to the community, and I hope to see it published in a different journal.

Minor issues (numbering by the Authors):

1. Throughout the paper, the presentation of math can be improved. I encourage the authors to carefully implement standard typography practices. For example, variables in italic, vectors in bold, units in roman font type. 'Dots' and 'crosses' should be used with caution, as they can be misinterpreted as cross products and dot products. Here is a reference for the IUPAC standard: <https://iupac.org/cms/wp-content/uploads/2016/01/ICTNS-On-the-use-of-italic-and-roman-fonts-for-symbols-in-scientific-text.pdf> In addition, the authors should define every variable where it is used (either in main text or figure caption). At several instances it is not clear what the mathematical formulation means unless one reads the SI (e.g. $S_{xv} = 0$ in Fig. 1b).

Thank you for drawing attention to this, we have revised the phrasing and added explanations of variables where applicable.

2. At several instances in the paper the range of the plot is too large, so that data cannot be assessed. E.g. Fig. 1C&D, the authors should show the regime of data, i.e. Ribosome fraction from 0 to 0.3 or proteome mass fraction from 0 to 0.2. In Fig. 2A there is a similar issue. It is impossible to judge if the model is a good or a bad fit.

We apologize and provide updated Fig 1 and Fig 2 in the manuscript.

3. In Fig. 4d, the authors make the statement that proteome fractions change linearly, but the reader can only judge the most abundant proteins. This data should be presented better. One option could be normalizing data.

We thank the Reviewer for this suggestion, and combining it with a related suggestion from Reviewer #2 (Point #4), we updated the Figure 4d.

4. "The maximal feasible growth rate that the model predicted can be linked directly to the dilution rate in the chemostat, allowing comparison of model prediction and data (Fig. 2a). " This panel is missing. Also, it would be great to know what this sentence is referring to. Does the model growth rate in a chemostat match the dilution rate?

Thank you for your attention to details, we rephrased the sentence as follows:

"The maximal feasible growth rate that the model predicted can be linked directly to the dilution rate in the chemostat (equal to the specific growth rate of the cell culture), allowing comparison of model prediction and experimental data from chemostats (Fig. 2a, 2c-d)."

5. "The (residual) glucose concentrations were calculated from documented (high) affinity of the transporters, which is close to 1 mM".

I did not understand this sentence. Please explain how that was done.

Our apologies for not providing detailed information on how we computed the residual glucose from the model simulations; the formula is \$[Glc_{Res}] = \frac{f_{sat}}{1-f_{sat}} \times K_M\$, with \$f_{sat}\$ being the saturation factor of the glucose transporters, \$K_M = 1 \text{ mM}\$, as mentioned in the text.

We appended the Supplementary Notes, Section “Growth-related Constraint 3: Growth media composition (exchange reactions)” with a paragraph referring to the calculation:

“In most of the experiments we performed ourselves and collected from literature sources, glucose was used as the sole (or main, in case of rich SC/YPD media) carbon source. Thus, assessment of residual glucose levels are an important consideration on correctly predicting metabolic behaviour. We computed the concentration of residual glucose from the model simulations, using formula $[Glc_{Res}] = \frac{f_{sat}}{1-f_{sat}} \times K_M$, with f_{sat} being the saturation factor of the glucose transporters and $K_M = 1 \text{ mM}$ being the Michaelis constant of high-affinity glucose transporters. Predictions of the residual glucose levels agree to the experimental observations that suggest only minute amounts of glucose in spent media from glucose-limited chemostats (see Fig 2A of the main text).”

6. The resulting relationship between growth rate and residual glucose concentration fit experimental data very well (Fig. 2b), validating our expectation that we could ignore intracellular glucose³².

In batch growth, where there is excess glucose, wouldn't we expect that growth rate is independent of glucose concentration? Does this only refer to the chemostat? Also, this seems to refer to Fig. 2A.

Thank you for spotting the reference to the wrong panel (Fig 2A instead of Fig 2B). In this part of the manuscript, we indeed referred to the glucose-limited conditions only. When the saturation of the transporter approaches 1 (batch conditions), the growth rate increase levels off, indeed, and becomes independent of glucose, as the Reviewer wonders. We actually made this point quite explicitly, as one of the interesting results that we highlight, is that the model indicates that at maximal saturation of the glucose transporter, its expression is (just) not actively limiting growth anymore. This indicates that *S. cerevisiae* expresses just enough transporter at glucose excess, and this becomes limiting at low glucose levels in the chemostat (as validated by increased glucose transporter expression beyond wild type during chemostat evolutionary experiments).

We made the reference to chemostat more explicit in the manuscript: additional text, with an accent on glucose levels having little-to-no influence to the batch cultures is provided in the response to Minor Point #7.

How about plotting log(glucose concentration) versus growth rate? This would make the plot resemble a typical enzyme kinetics plot and could be easier to read.

We have here used the plot commonly used by experts in the field of chemostat cell physiology (such as our partners in Delft). It is in fact a well-known Monod curve (with high affinity) but with the axis swapped to reflect the control parameter on the x-axis.

We made a comment in the legend to make it explicit:

“Fig 2. <...> . a. ... Note that this resembles a Monod growth curve but with the dependent and independent axis swapped, as we control growth rate in a chemostat.”

And how do the authors conclude that they can ignore internal glucose?

We apologize for being too brief here. What we meant was that, since we can describe the data very well with an irreversible Michaelis-Menten equation, we conclude that we can safely neglect the potential inhibitory effects of intracellular glucose. In glucose-limited chemostats, the concentration of intracellular glucose is expected to be very low and the effect therefore very small. A strong impact of intracellular glucose would only complicate the mapping between saturation level of the transporter and the extracellular glucose concentration, but not the effect of transport saturation on growth rate and metabolic strategies.

We provide this explanation in the Supplementary Notes (Section “Growth-related Constraint 3: Growth media composition (exchange reactions)”):

“We modeled glucose uptake as irreversible process and, as a result, obtained good agreement with the experimental data on both the residual glucose concentrations and glucose uptake rates. Therefore, we conclude that we can safely neglect the potential inhibitory effects of intracellular glucose concentration. In glucose-limited chemostats, the concentration of intracellular glucose is expected to be very low and the effect therefore very small. It should be noted that a strong impact of intracellular glucose would only complicate the mapping between saturation level of the transporter and the extracellular glucose concentration, but not the effect of transport saturation on growth rate and metabolic strategies.”

To which we refer in the main text:

“The resulting relationship between growth rate and residual glucose concentration fit experimental data very well (Fig. 2a), validating our expectation that we could ignore **glucose efflux from the cells due to minute levels of intracellular glucose**³² (see Supplementary Notes for details).”

7. A discussion of Fig. 2B is missing.

Thank you. In line with this suggestion, we included a short comment in the text on saturation factor-growth rate dependence:

“<...> very well (Fig. 2a), validating our expectation that we could ignore intracellular glucose³². **Increasing glucose transporter saturation increased predicted growth rate, and the effect saturated (Fig. 2b), suggesting that at maximal growth rate further increase in glucose availability has little impact.** Predicted biomass yield (Fig. 2c) and fluxes (Fig. 2d) corresponded <...>”

8. In Fig. S1 it is not clear what ‘relative’ is referring to.

We apologize for the incomplete caption; the ‘relative’ refers to relative flux to that of hexokinase reaction. We appended the caption of Fig S1, stating this explicitly:

“**FigS1.** Comparison of experimentally determined (abscissa) and predicted (ordinate) fluxes in central carbon metabolism of glucose-limited *S. cerevisiae*. Simulations of glucose-limited growth were performed by varying the hexose transporter saturation, as for the Figure 2d of the main text. Experimental data for the intracellular fluxes at three dilution rates ($D = 0.1, 0.3$ and $0.4 h^{-1}$) were taken from (Frick & Wittmann, 2005). **Flux values were normalized to those of the hexokinase reaction.**”

9. What are the lines in Fig. 5B?

In Fig. 5b, dots represent experimental flux measurements, while lines correspond to the model predictions. We had this information included at the end of the caption of the Figure 5, but made this more clear by changing the caption as follows:

Fig. 5 The effect of translation inhibition by cyclohexamide on growth rate, fluxes and proteome fractions in controlled aerobic batch fermentations on glucose. **a.** Dependency of culture optical density (OD) on the time post-inoculation to the medium supplemented with cycloheximide. Lines are values of consecutive OD measurements, points represent the times when cultures were sampled. ~~For b-d~~ **b.** Comparison of pcYeast predictions and experimental data; lines are model predictions; symbols are experimental data points. **b.** Main catabolic fluxes as a function of the growth rate. **c.** Ribosomal proteome fractions. Data from Fig. 1c are included for comparison. **d.** Proteome fractions measured for key metabolic pathways, and the minimal proteome fractions predicted by pcYeast. ~~For b-d, lines are model predictions; symbols are experimental data points.~~

Reviewers' Comments:

Reviewer #1:

Remarks to the Author:

I had pointed out in my previous review, I had problems with the general flow of the article, and the authors have made it better.

Going through the reviews of other referees, I feel that the authors have made a decent effort at answering their queries.

Although there are assumptions and shortcomings about those assumptions (such as the mitochondrial calculations, unspecified proteins, etc.) which can limit the predictive power and generality of the model, one has to start somewhere and I am in favor of at least trying the hands at 'a model'. Therefore, I feel that the authors, despite the shortcomings of their model, make a good effort with known literature, to construct such a model and compare its predictions to experiments.

Reviewer #2:

Remarks to the Author:

The authors adequately addressed the clarifying statements I suggested in my first review. Therefore, I am satisfied with their responses to my comments.

I also had a chance to examine the comments of the other two reviewers and the detailed responses of the authors. While Reviewer 3 offered some really useful comments about the tuning and use of certain model parameters, they seem to be uncomfortable with the idea of modeling with incomplete and uncertain information. However, this is the essence of biological systems modeling since there also will be MUCH less data than needed to fully parameterize the models and many parameters always will be HIGHLY uncertain. I believe that the authors have done a good job of addressing the comments of Reviewer 3 to the extent possible given current knowledge. Perhaps one small change that could be useful is to distinguish those predictions that are built into the model versus those that are truly emergent.

Reviewer #3:

Remarks to the Author:

Review of the revised paper

The authors have done a great job improving the paper and answering my questions. In fact, my previous review may have been too harsh. The authors are right in defending that they have been transparent about their modeling in their manuscript, and the revised version is certainly as clear and transparent as it can get.

It would have been great to see the theoretical prediction, when which compartment becomes limiting to see on a stronger experimental or theoretical footing, but this paper is a good step forward and I believe it will be appreciated by the field.

Below we provide the responses to the final remarks of the Reviewers for this manuscript (Author responses in blue).

Reviewer #1 (Remarks to the Author):

I had pointed out in my previous review, I had problems with the general flow of the article, and the authors have made it better.

Going through the reviews of other referees, I feel that the authors have made a decent effort at answering their queries.

We appreciate the positive evaluation of both our responses and improvements towards better readability of the article.

Although there are assumptions and shortcomings about those assumptions (such as the mitochondrial calculations, unspecified proteins, etc.) which can limit the predictive power and generality of the model, one has to start somewhere and I am in favor of at least trying the hands at 'a model'. Therefore, I feel that the authors, despite the shortcomings of their model, make a good effort with known literature, to construct such a model and compare its predictions to experiments.

Thank you for this comment. We indeed tried to collect as much information as possible to increase the predictive power of the model, yet, as Reviewers #1 and #2 rightfully indicate, uncertainty is always present at a certain degree. We welcome Reviewer's commitment to "<...> at least try[ing] the hands at 'a model'", as computational modelling of complex biological systems is a continuous and iterative effort.

Reviewer #2 (Remarks to the Author):

The authors adequately addressed the clarifying statements I suggested in my first review. Therefore, I am satisfied with their responses to my comments.

We are very happy to hear this.

I also had a chance to examine the comments of the other two reviewers and the detailed responses of the authors. While Reviewer 3 offered some really useful comments about the tuning and use of certain model parameters, they seem to be uncomfortable with the idea of modeling with incomplete and uncertain information. However, this is the essence of biological systems modeling since there also will be MUCH less data than needed to fully parameterize the models and many parameters always will be HIGHLY uncertain. I believe that the authors have done a good job of addressing the comments of Reviewer 3 to the extent possible given current knowledge.

Thank you for assessing our responses to the comments of other Reviewers, and for the positive perception.

Perhaps one small change that could be useful is to distinguish those predictions that are built into the model versus those that are truly emergent.

Thank you for this suggestion. We sought to highlight the nature of predictions (coming from calibration or as a genuine prediction) by putting different results into separate sections of the manuscript. Yet, with the help of the Editorial Office, will try to modify the phrasing where applicable to make this more explicit.

Reviewer #3 (Remarks to the Author):

Review of the revised paper

The authors have done a great job improving the paper and answering my questions. In fact, my previous review may have been too harsh. The authors are right in defending that they have been transparent about their modeling in their manuscript, and the revised version is certainly as clear and transparent as it can get.

We really appreciate the Reviewer's feedback and willingness to accept our position. The reviewer's questions indeed led us to improving the manuscript a lot. This is how review should be, thank you for this.

It would have been great to see the theoretical prediction, when which compartment becomes limiting to see on a stronger experimental or theoretical footing, but this paper is a good step forward and I believe it will be appreciated by the field.

Thank you for the suggestion. We agree and are indeed continuing the iterative process of model prediction and validation with experimental data, to improve the model and to increase its scope, and we hope this paper is only a first good step forward.